# A WDR35-dependent coat protein complex transports ciliary membrane cargo vesicles to cilia

Tooba Quidwai[1†], Jiaolong Wang[2], Emma A Hall[1], Narcis A Petriman[2], Weihua Leng[3], Petra Kiesel[3], Jonathan N Wells[1], Laura C Murphy[1], Margaret A Keighren[1], Joseph A Marsh[1], Esben Lorentzen[2], Gaia Pigino[3,4], Pleasantine Mill[1]*

[1]MRC Human Genetics Unit, Institute of Genetics and Cancer, University of Edinburgh, Edinburgh, United Kingdom; [2]Department of Molecular Biology and Genetics, Aarhus University, Aarhus, Denmark; [3]Max Planck Institute of Molecular Cell Biology and Genetics, Dresden, Germany; [4]Human Technopole, Milan, Italy

**Abstract** Intraflagellar transport (IFT) is a highly conserved mechanism for motor-driven transport of cargo within cilia, but how this cargo is selectively transported to cilia is unclear. WDR35/IFT121 is a component of the IFT-A complex best known for its role in ciliary retrograde transport. In the absence of WDR35, small mutant cilia form but fail to enrich in diverse classes of ciliary membrane proteins. In *Wdr35* mouse mutants, the non-core IFT-A components are degraded and core components accumulate at the ciliary base. We reveal deep sequence homology of WDR35 and other IFT-A subunits to α and ß' COPI coatomer subunits and demonstrate an accumulation of 'coat-less' vesicles that fail to fuse with *Wdr35* mutant cilia. We determine that recombinant non-core IFT-As can bind directly to lipids and provide the first in situ evidence of a novel coat function for WDR35, likely with other IFT-A proteins, in delivering ciliary membrane cargo necessary for cilia elongation.

*For correspondence:
pleasantine.mill@igmm.ed.ac.uk

Present address: †UMass Chan Medical School, Worcester, United States

Competing interest: The authors declare that no competing interests exist.

## Editor's evaluation

This work supports a model in which the intraflagellar transport complex, IFT-A, acts as a coat to support Golgi to cilia traffic of ciliary cargos. Data also show that proteins in the IFT-A complex are co-dependent and bind to specific lipids, consistent with such a coat function. Consistent with this model, in the absence of one core component of IFT-A, IFT122/WDR35, uncoated vesicles accumulate at the base of cilia, suggesting a requirement for the coat in fusion at cilia.

## Introduction

The primary cilium is a highly specialized sensory organelle and signaling hub compartmentalized from the rest of the cell and positioned with a unique interface towards the extracellular environment. Analogous to a cell's antenna, many roles for cilia have emerged in development, disease, and homeostasis (*Reiter and Leroux, 2017*). Enrichment of signaling receptors and effectors in ciliary membranes is critical for cilia function, yet all biosynthesis of cilia-localized membrane proteins occurs in the endoplasmic reticulum and is sorted by the Golgi and vesicular membrane traffic system to efficiently route cargo-laden vesicles for incorporation into the elongating ciliary membrane. The details of this highly efficient, directed transport process for the delivery of diverse cargos to cilia remain unclear.

**eLife digest** Most human cells have at least one small hair-like structure on their surface called a cilium. These structures can act as antennae and allow the cell to sense signals from the rest of the body. To do this, they contain proteins that differ from the rest of the cell. The content of cilia depends on regulated delivery of these proteins in and out of cilia by a process called the intraflagellar transport or IFT, which involves a large complex made of several proteins. This complex shuttles the cargo proteins back and forth between the base and the tip of the cilia.

However, ciliary proteins are not produced in the cilia; instead, they are made in a different part of the cell and then they are transported to the ciliary base. At the point where they enter the cilia, they were thought to bind to the assembling IFT 'trains' and be transported across the ciliary gate to the positions where they are needed in cilia. One of the components of the IFT machinery is a protein called WDR35, also known as IFT121. If the gene that codes for this protein is faulty or missing, it results in severe disorders in both humans and mice including a range of potentially lethal skeletal dysplasias.

Interestingly, without WDR35, cells cannot build functional cilia. The absence of this protein not only disrupts IFT, stopping certain ciliary proteins and their associated membranes from entering cilia; it also causes a 'traffic jam' with a pile-up of transport intermediates from the place in cell where they are made to the cilia. It is unclear why a mutation in one of the components of the IFT would have this effect, raising the question of whether WDR35, or IFTs a whole, has another role in bringing the cargo proteins into the cilia.

To understand this phenomenon, Quidwai et al. analysed the structure of WDR35 and other IFT proteins and found that they are very similar to a protein complex called COPI, which is involved in transporting membrane proteins around the cell. When certain proteins are newly made, they are stored in small lipid bubbles – called vesicles – that then selectively move to where the proteins are needed. COPI coats these vesicles, helping them get to where they need to go in a process called vesicular transport. Quidwai et al. found that WDR35 and other IFT proteins are able to bind to specific types of lipid molecules, suggesting that they might be assisting in a form of vesicle transport too. Indeed, when mouse cells grown in the lab were genetically engineered so they could not produce WDR35, coatless vesicles accumulated around the base of the cilia. Adding back WDR35 to these mutant cells rescued these defects in vesicle transport to cilia as well as allowed functional cilia to be formed.

These results provide evidence that WDR35, likely with other IFT proteins, acts as a COPI-like complex to deliver proteins to growing cilia. Further research will investigate the composition of these vesicles that transport proteins to cilia, and help pinpoint where they originate. Quidwai et al.'s findings not only shed light on how different genetic mutations found in patients with cilia dysfunction affect different steps of transporting proteins to and within cilia. They also increase our understanding of the cellular roadmap by which cells shuttle building blocks around in order to assemble these important 'antennae'.

In mammalian cells, electron microscopy (EM) studies reveal the Golgi stacks closely apposed to the mother centriole (*Sorokin, 1962*; *Wheatley, 1969*). During intracellular ciliogenesis, small vesicles are recruited, most likely from the Golgi, to the mother centriole, where they fuse to form a large preciliary vesicle (PCV) attached at the distal appendages (*Yee and Reiter, 2015*). More secondary vesicles later fuse with the PCV, allowing elongation of cilia. Interestingly, the Golgi remains close to mature cilia, suggesting a continuous exchange of materials, enabling cilia maintenance (*Sorokin, 1962*; *Wheatley, 1969*). Several proteins essential for ciliogenesis localize to both the Golgi and the mother centriole and are implicated in this early stage of ciliogenesis including CCDC41 (CEP83), IFT20, HOOK2, and CEP164 (*Baron Gaillard et al., 2011*; *Follit et al., 2006*; *Graser et al., 2007*; *Joo et al., 2013*; *Schmidt et al., 2012*; *Tanos et al., 2013*). In some cases, including HOOK2 and CEP164, these components recruit Rab8a and Rabin-8, which facilitate membrane transport to cilia (*Baron Gaillard et al., 2011*; *Moritz et al., 2001*; *Nachury et al., 2007*). For some specific ciliary cargos, including rhodopsin (*Wang and Deretic, 2014*) and PKD2 (*Follit et al., 2008*; *Follit et al., 2006*; *Hoffmeister et al., 2011*; *Kim et al., 2014*; *Noda et al., 2016*), Golgi-to-cilia transport mechanisms have

been described. However, these processes seem to involve cargo-specific traffic modules. A more universal Golgi-to-cilia transport machinery, if one exists, has yet to be identified.

In contrast to traffic to cilia, movement of cargos within cilia requires highly conserved motor-driven macromolecular cargo binding complexes that traffic along axonemal microtubules closely apposed against the ciliary membrane, in a process known as intraflagellar transport (IFT) (*Cole, 2009*; *Kozminski et al., 1993*; *Pazour et al., 1998*; *Pigino et al., 2009*; *Rogowski et al., 2013*; *Rosenbaum and Witman, 2002*). Bidirectional movement of IFT complexes regulates cilia content; the IFT-B complex aids in kinesin-dependent anterograde transport of cargo, whilst the IFT-A complex is required for retrograde transport driven by dynein motors (*Blacque et al., 2006*; *Efimenko et al., 2006*; *Jonassen et al., 2012*; *Lee et al., 2008*; *Piperno et al., 1998*; *Tran et al., 2008*; *Tsao and Gorovsky, 2008*). The IFT-A complex is composed of three core (IFT144/WDR19, IFT140/WDTC2, IFT122/WDR10) and three non-core proteins (IFT139/TTC21B/THM1, IFT121/WDR35, and IFT43) (*Behal et al., 2012*; *Hirano et al., 2017*; *Piperno et al., 1998*). However, beyond classical retrograde ciliary traffic defects (an inappropriate accumulation of cargos within the cilium) mutations in *IFT144*, *IFT140, IFT122, IFT121/WDR35*, and *IFT43* result in either severe reduction in cilia length or complete loss of cilia, implying they also have critical roles in transport of cargo to cilia (*Avidor-Reiss et al., 2004*; *Caparrós-Martín et al., 2015*; *Duran et al., 2017*; *Hirano et al., 2017*; *Liem et al., 2012*; *Mill et al., 2011*; *Takahara et al., 2018*; *Zhu et al., 2017*). Indeed, several IFT-A mutants fail to localize a range of ciliary membrane proteins including EVC1/2, SMO, ARL13B, INPP5E, and SSTR3 to cilia (*Brear et al., 2014*; *Caparrós-Martín et al., 2015*; *Fu et al., 2016*; *Hirano et al., 2017*; *Jensen et al., 2010*; *Lee et al., 2008*; *Liem et al., 2012*; *Mukhopadhyay et al., 2010*; *Takahara et al., 2018*). However, the mechanism of transport and the location of any IFT-A extra-ciliary function remain unclear.

The movement of cargos between membranes of spatially separated organelles in the cytoplasm involves vesicular traffic. Indeed, IFT proteins have been observed to localize to various endomembranes and vesicular compartments outside cilia. For example, the IFT-B protein IFT20 localizes to the Golgi (*Follit et al., 2006*; *Noda et al., 2016*), whereas both IFT-B (IFT20, IFT27, IFT46, IFT52, IFT57, IFT88, and IFT172) and IFT-A proteins (IFT139, IFT140) cluster around periciliary vesicles, shown by immuno-EM and light microscopy (*Sedmak and Wolfrum, 2010*; *Wood et al., 2012*; *Wood and Rosenbaum, 2014*). Direct interaction of IFTs with membranes in vitro has also been described where the adaptor TULP3 and phosphoinositides mediate the membrane association of IFT-As (*Mukhopadhyay et al., 2010*). More recently, purified IFT172 was shown to bind to lipids and pinch off smaller vesicles, similar in size to classic COPI vesicles (*Wang et al., 2018*). It has been postulated that IFT proteins have evolved from membrane traffic coat complexes: soluble multimeric protein complexes that 'coat' donor membranes, facilitating cargo enrichment and membrane remodeling prior to traffic and fusion with target membranes (*Jékely and Arendt, 2006*; *van Dam et al., 2013*). Nonetheless, a functional requirement for an IFT-dependent vesicle-to-cilia traffic module and what its dynamic architecture may resemble is currently unknown.

To dissect how traffic of newly synthesized ciliary membrane proteins to the cilium occurs, we undertook a series of biochemical and imaging experiments in *Wdr35* null mouse embryonic fibroblasts (MEFs) (*Caparrós-Martín et al., 2015*; *Mill et al., 2011*). To distinguish extra-ciliary functions from canonical retrograde traffic defects, we compared *Wdr35*$^{-/-}$ phenotypes with those of the retrograde IFT dynein *Dync2h1*$^{-/-}$ (*Criswell et al., 1996*; *Huangfu and Anderson, 2005*; *Porter et al., 1999*; *Signor et al., 1999*). Whilst accumulations of intact IFT-B proteins were observed inside cilia in both mutants, only in the absence of WDR35 does the IFT-A holocomplex fragment and fail to enter *Wdr35*$^{-/-}$ cilia. Without intact IFT-A, we observe broad defects in the ciliary import of diverse membrane and lipidated proteins, as well as an accumulation of 'coat-less' vesicles around the base of *Wdr35* mutants, which fail to fuse with the ciliary sheath. We demonstrate that together recombinant non-core IFT-A proteins (WDR35, IFT43, and IFT139) are sufficient to specifically bind lipids in vitro. Together with our localization data, our results provide the first in situ evidence of a WDR35-dependent coat required to deliver essential cargo from vesicles to cilia.

## Results

### *Wdr35* null cells have rudimentary, short cilia with intact transition zones

We utilized primary MEFs carrying null mutations in two components of the retrograde IFT machinery (*Figure 1E*), one part of the motor complex that moves cargos (the retrograde dynein heavy chain *Dync2h1*), and the non-core IFT-A component *Wdr35,* in order to dissect the stage at which ciliogenesis defects occurred (*Caparrós-Martín et al., 2015*; *Mill et al., 2011*). Cilia length measured by acetylated α-tubulin staining was drastically reduced in both *Wdr35*$^{-/-}$ (0.48 μm mean ± 0.35 SD) and *Dync2h1*$^{-/-}$ (0.76 μm mean ± 0.35 SD) mutants compared to wild type (WT) (2 μm mean ± 0.45 SD) MEFs (*Figure 1A and B*). Given there was no reduction in cilia number (*Figure 1C*), as previously shown (*Fu et al., 2016*; *Liem et al., 2012*; *Mukhopadhyay et al., 2010*), our results suggest that DYNC2H1 and WDR35 are needed for cilia elongation at later stages of ciliogenesis. Defects in centriolar satellite traffic, implicated in ciliogenesis, were previously reported for *WDR35* mutant RPE-1 cells (*Fu et al., 2016*); however, we saw no difference in levels or localization of endogenously tagged PCM1 protein (PCM1-SNAP), which marks centriolar satellites in MEFs (*Figure 1—figure supplement 1*, *Figure 1—video 1*). In *Caenorhabditis elegans* non-core IFT-A mutants, extension of the MKS module into the axoneme from the transition zone due to failure of cargo retrieval had been reported (*Scheidel and Blacque, 2018*). However, we observed intact transition zone modules as shown by NPHP1 and MKS1 localization in both mammalian mutants (*Figure 1D and E*). We noted that *Wdr35*$^{-/-}$ axonemes, while acetylated, were not polyglutamylated, suggesting differences in tubulin post-translational modifications (PTMs) and stability (*Figure 1D*). Together, these data suggest that the initial steps of ciliogenesis occur in both *Dync2h1*$^{-/-}$ and *Wdr35*$^{-/-}$ mutants; however, subsequent axoneme elongation may be differentially affected.

### *Wdr35* null cells have intact IFT-B complexes with a retrograde defect and unstable IFT-A holocomplexes that fail to enter cilia

Axoneme elongation during cilia assembly requires the import of key building blocks from their place of synthesis in the cell body into the cilium across the transition zone via IFT. We focused first on the anterograde, IFT-B machinery, monitoring two subunits IFT81 and IFT88. We found that IFT-B complex proteins have similar retrograde traffic defects in both *Wdr35*$^{-/-}$ and *Dync2h1*$^{-/-}$ cells (*Figure 2A and B*), accumulating beyond the length of the acetylated axoneme. We next looked to see if IFT-B complex assembly is disturbed in the absence of WDR35 by immunoprecipitation (IP) of endogenous IFT88, followed by mass spectrometry (MS) to identify co-purifying subunits. IFT88 is the link between IFT-B1 and IFT-B2 complexes (*Figure 1E*), interacting with IFT38 on the IFT-B2 side and IFT52 on the IFT-B1 side (*Katoh et al., 2016*; *Mourão et al., 2016*; *Taschner et al., 2016*). MS analysis of immunoprecipitates from E11.5 *Wdr35*$^{+/+}$ and *Wdr35*$^{-/-}$ embryo lysates revealed no statistically significant differences in stoichiometric composition of IFT-B complexes (*Figure 2C and D*). We were able to isolate almost the entire IFT-B complex (14 out of 16 IFT-B components) aside from IFT70, which is not yet reported in mouse as well as IFT25, which binds IFT27 to form a heterodimer (*Bhogaraju et al., 2011*; *Funabashi et al., 2017*; *Katoh et al., 2016*; *Wang et al., 2009*) and is necessary for Hh signaling (*Keady et al., 2012*). Because the composition of the IFT-B complex and its ability to enter cilia each appear unaltered, we conclude that exit from cilia is impaired in the absence of WDR35.

We next examined the composition of the IFT-A holocomplex in WT vs. *Wdr35*$^{-/-}$ embryos by IP of endogenous IFT-A core protein IFT-140 and its interactors. Whilst IFT140 immunoprecipitated all six components of the IFT-A complex from *Wdr35*$^{+/+}$ embryo lysates, in *Wdr35*$^{-/-}$ samples both non-core components IFT139 and IFT43 were missing from our MS datasets (*Figure 3A*). Their absence was confirmed by immunoblotting (*Figure 3B*). Moreover, core components IFT122 and IFT144 were also significantly reduced in the purified mutant complex (*Figure 3A*), suggesting that WDR35 is critical for the stability of the IFT-A complex and its components. We also compared total cellular levels of IFT-A component proteins and found that IFT139 and IFT43 levels were also undetectable on blots with lysates from both *Wdr35*$^{-/-}$ embryos (*Figure 3C and D*) and MEFs (*Figure 3—figure supplement 1A and B*). This suggests that WDR35 is not only critical for the formation of stable IFT-A holocomplex but is also required for stability of its non-core components.

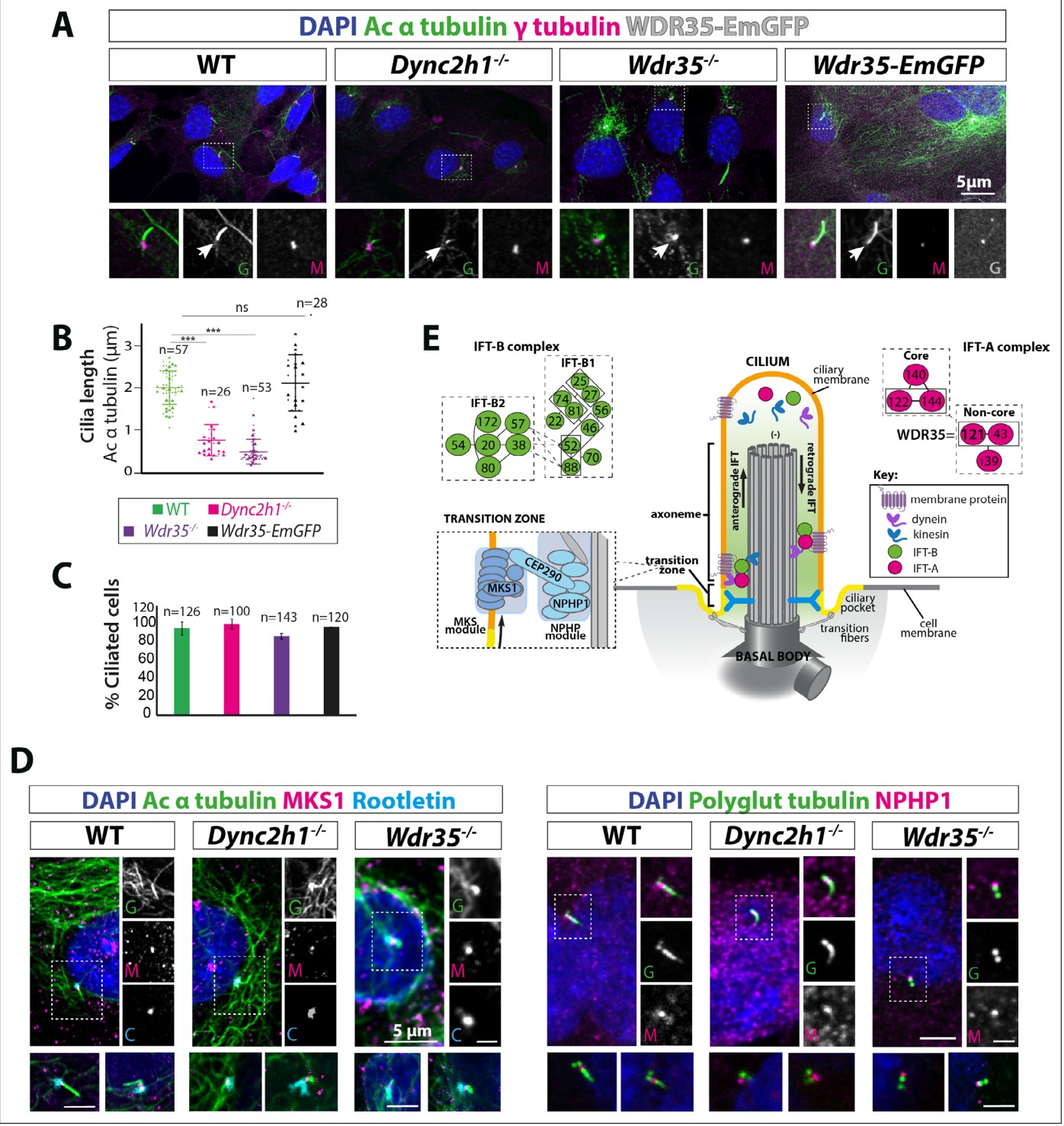

**Figure 1.** *Wdr35⁻ᐟ⁻* and *Dync2h1⁻ᐟ⁻* mutant cells have a drastic reduction in cilia length but have no difference in the number of cilia. (**A**) Wild type (WT) and mutant mouse embryonic fibroblasts (MEFs) and those rescued by transiently expressing WDR35-EmGFP serum-starved for 24 hr, fixed and stained with acetylated α-tubulin (green) and γ-tubulin (magenta), nuclei (blue). Boxed regions are enlarged below, and arrows point at ciliary axoneme stained for acetylated α-tubulin. (**B**) Quantification of cilia length for acetylated α-tubulin. n = total number of cells from three different biological replicates (represented by different shapes). Asterisk denotes significant p-value from t-test: *$p < 0.05$, **$p < 0.01$, ***$p < 0.001$. (**C**) Percentage of acetylated α-tubulin-positive ciliated cells. (**D**) 24 hr serum-starved WT and mutant MEFs stained for nuclei (blue), acetylated α-tubulin/polyglutamylated tubulin (green), rootletin (cyan), and transition zone proteins MKS1/NPHP-1 (magenta) show no difference in the localization of transition zone proteins MKS1 and NPHP-1. Gray scale enlarged regions are labeled green (G), magenta (M), and cyan (C). (**E**) Schematic of intraflagellar transport (IFT) pathway in cilia.

*Figure 1 continued on next page*

*Figure 1 continued*

The online version of this article includes the following video and figure supplement(s) for figure 1:

**Source data 1.** Full data points and stats test of cilia length for *Figure 1B*.

**Figure supplement 1.** The organization of centriolar satellites (CS) around *Wdr35*[-/-] cilia is not changed.

**Figure 1—video 1.** The organization of centriolar satellites is not disrupted in *Wdr35*[-/-] mutants.

https://elifesciences.org/articles/69786/figures#fig1video1

In contrast, the individual core components of the IFT-A complex were nearly equally expressed in WT and *Wdr35*[-/-] lysates, except for IFT122, which had higher expression levels in *Wdr35*[-/-] MEFs (*Figure 3C and D*, *Figure 3—figure supplement 1A and B*). Other core components have been shown to have higher levels in the absence of WDR35 in human fibroblasts (*Duran et al., 2017*). Thus, our work also supports previous studies demonstrating a level of interdependence in the levels of IFT-A subunits, which might be required for their coordinated function (*Behal and Cole, 2013*; *Duran et al., 2017*; *Fu et al., 2016*; *Picariello et al., 2019*; *Zhu et al., 2017*).

These results suggest that WDR35 might be a link between IFT-A core and non-core proteins, which when absent results in the decreased abundance of IFT-A non-core subunits. To further distinguish between increased protein degradation from transcriptional changes, control and mutant MEFs were treated with the proteasome inhibitor MG-132 (20 µM) (*Figure 3E*). Treated cells displayed increased levels of IFT43, which suggests that in the absence of WDR35, non-core proteins may be targeted by the proteasomal degradation pathway. Interestingly, IFT139 and IFT121 are degraded in *IFT43* null cells and both are rescued similarly by MG-132 treatment (*Zhu et al., 2017*), confirming that the stability of IFT-A complex proteins is interdependent.

We next looked at the localization and levels of the IFT-A components by immunofluorescence. IFT-A components were present in *Dync2h1*[-/-] cilia, suggesting that entry of IFT-A holocomplexes is not affected, but return from the distal tip is compromised in the absence of the dynein motor (*Figure 3F*, *Figure 3—figure supplement 1C*). In contrast, in *Wdr35*[-/-] MEFs, IFT-A core components fail to enter cilia and remain restricted at the ciliary base (*Figure 3F*), as shown by the difference in length covered by IFT-A components relative to cilia length measured by acetylated tubulin staining (*Figure 3—figure supplement 1C*), whereas non-core proteins were undetectable, consistent with degradation (*Figure 3C–F*). These results are consistent with previous reports of the interdependence of IFT-A components for ciliary localization. IFT140 is decreased in cilia of *IFT122* mutants in mouse and fly (*Lee et al., 2008*; *Qin et al., 2011*), IFT139 is reduced in the flagella of *Chlamydomonas* with *IFT144* mutation (*Iomini et al., 2009*), and IFT144 fails to get recruited into cilia in *WDR35*[-/-] RPE cells (*Fu et al., 2016*). IFT-A proteins require CPLANE chaperones for holocomplex assembly and cilia entry (*Toriyama et al., 2016*). In all cases, failure of IFT-A holocomplex integrity impairs its recruitment into the cilia axoneme. Recent cryo-EM work had suggested that IFT-A is being carried by IFT-B trains inside the *Chlamydomonas* flagella in WT cells and these structures are missing in the *IFT139* mutant (*Jordan et al., 2018*). Our work in the mammalian system in the absence of WDR35 has a similar effect with IFT-B proteins accumulating inside the cilium whilst IFT-A core proteins accumulate proximal to the cilia base, and the non-core components are degraded.

## Membrane proteins fail to be recruited into *Wdr35*[-/-] cilia

Cilia membrane protein cargos are synthesized in the cell body (rough ER) and traffic into cilia through a variety of direct and indirect routes. These include lateral diffusion from the plasma membrane (*Leaf and Von Zastrow, 2015*; *Milenkovic et al., 2009*), recycling of plasma membrane proteins via the endocytic pathway (*Boehlke et al., 2010*), as well as more directly from Golgi-derived vesicles (*Follit et al., 2008*; *Follit et al., 2006*; *Kim et al., 2014*; *Witzgall, 2018*). Moreover, ciliary membrane content is dynamically regulated in response to external signals. First, we tested appropriate dynamic localization of the GPCR Smoothened (SMO), which is recruited to the cilia in response to Hh ligand (*Figure 4A*). SMO is already present in *Dync2h1*[-/-] mutant cilia, even in the absence of Hh. In contrast, even in the presence of Hh activation, SMO fails to enter *Wdr35*[-/-] cilia. We investigated endogenous levels and localizations of membrane-associated GTPases ARL13B and ARL3, which are enriched in cilia in control cells (*Figure 4A*). We saw that while they accumulate in excess in *Dync2h1*[-/-] mutants as per a retrograde defect, strikingly they fail to be recruited into *Wdr35*[-/-] cilia. Detecting low levels of

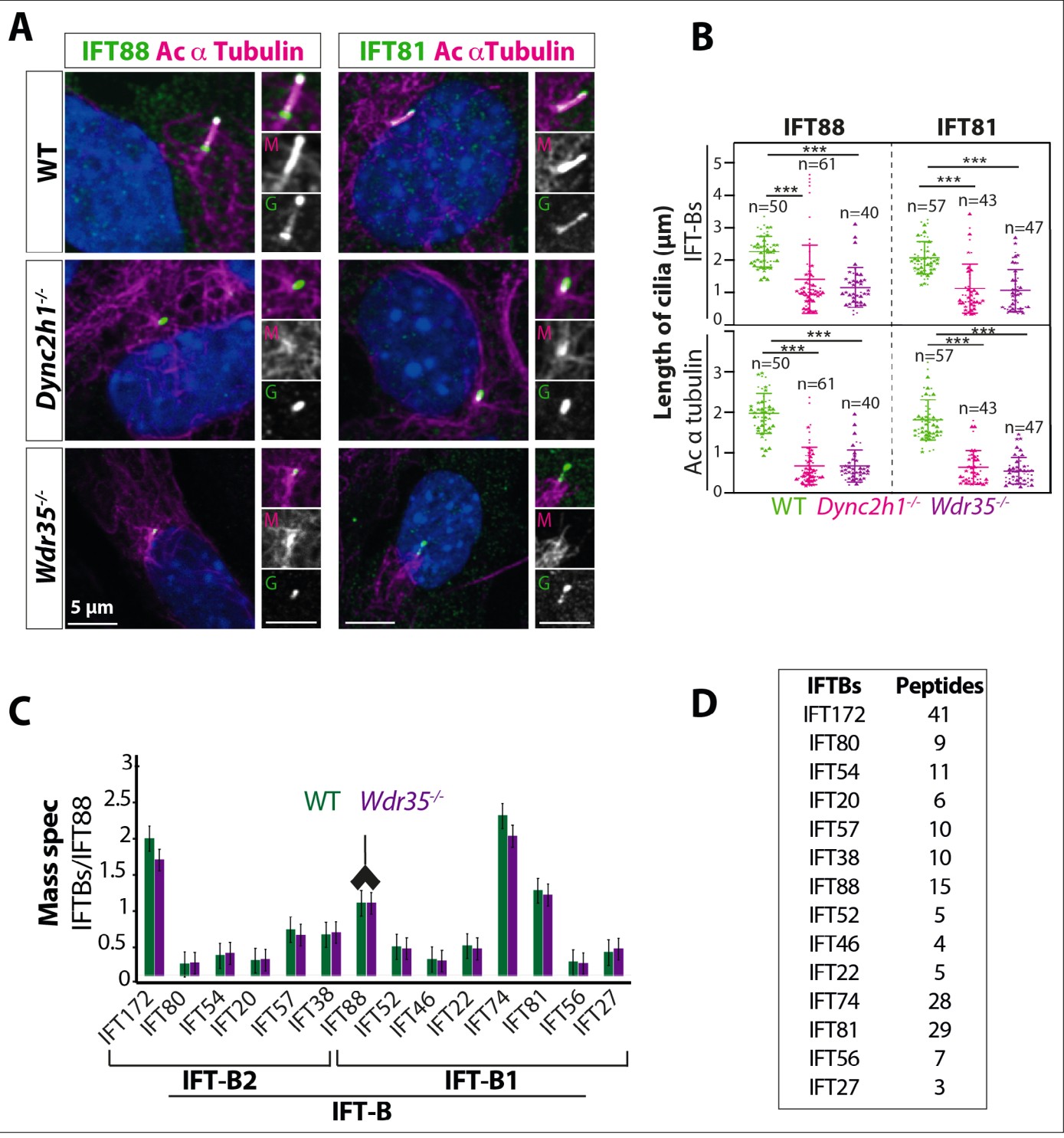

**Figure 2.** *Wdr35⁻/⁻* cilia exhibit retrograde transport defects of IFT-B, similar to *Dync2h1⁻/⁻*, although IFT-B complex assembly is unaffected. (**A**) IFT-B (green) accumulates beyond the axoneme (Ac-α-tubulin, magenta) in *Wdr35* and *Dync2h1* mutant cilia from 24 hr serum-starved and fixed mouse embryonic fibroblasts (MEFs). (**B**) Length quantification shows IFT-B accumulates beyond acetylated α-tubulin in significantly shorter mutant cilia. n = total number of cells from three different biological replicates represented by different shapes. Asterisk denotes significant p-value from t-test: *p<0.05, **p<0.01, ***p<0.001. Scale bars = 5 μm. (**C, D**) Despite differences in localization, IFT88 immunoprecipitation/mass spectrometry (IP/MS) analysis of E11.5 wild type (WT) and *Wdr35⁻/⁻* littermate embryos reveals no difference in the composition of the IFT-B complex. Antibody highlights bait (IFT88) for IP. (**C**) Normalized label-free quantification intensities (LFQs) to IFT88 intensity reveal no difference between WT and *Wdr35⁻/⁻* IFT-B complex composition. N = 4 embryos/genotype. (**D**) The number of unique peptides identified in IP/MS.

*Figure 2 continued on next page*

*Figure 2 continued*

The online version of this article includes the following figure supplement(s) for figure 2:

**Source data 1.** Full data points and stats test of IFT-B and cilia length (*Figure 2B*).

endogenous protein localization and their mislocalization in *Wdr35* mutants by immunofluorescence can be challenging. To overcome this, we transiently expressed membrane cargos, including fluorescently tagged SMO and ARL13B (*Figure 4B*, *Figure 4—video 1*), which effectively traffic into the cilia of WT cells. However, they fail to localize to *Wdr35$^{-/-}$* cilia, with some accumulation at the cilia base. Interestingly, in our *Wdr35$^{-/-}$* cells, SMO was predominantly localized to vesicles in the cytoplasm of mutant cells, whereas overexpressed ARL13B when not transported into cilia, is concentrated on other membranes, particularly the plasma membrane and pericentrosomal vesicles (*Figure 4B*, *Figure 4— video 1*).

In trypanosomes, localization of flagellar membrane proteins was shown to be dependent on lipid rafts highly enriched in axonemes (*Tyler et al., 2009*). Here, dual acylation was shown to direct potential association with lipid rafts, membrane microdomains that function as specialized platforms for protein/lipid transport and signaling. Indeed, ARL13B requires palmitoylation for its cilia membrane targeting and ciliogenesis in worms and mammals (*Cevik et al., 2010*; *Li et al., 2010*; *Roy et al., 2017*), where it acts as the cilia-localized GEF for ARL3, driving it to release lipid-modified cargos from carriers UNC119 and PDE6δ within cilia membranes (*Gotthardt et al., 2015*; *Kapoor et al., 2015*). As ARL13B and ARL3 fail to localize to mutant cilia, we next asked about the ability to recruit general lipidated cargo in *Wdr35$^{-/-}$* MEFs. We examined the localization of lipidated motifs tagged to EGFP (*Williams et al., 2014*) to look at specialized membrane microdomains. In WT MEFs, untagged EGFP is present in the cell, but not in the cilium. When tagged with either myristoylation and palmitoylation (MyrPalm) or dual palmitoylation (PalmPalm) motifs, EGFP robustly enriches within cilia (*Figure 4C*). We observed no enrichment of dual geranylation (GerGer) modified EGFP within control fibroblast primary cilia (data not shown) in contrast to the low-level expression previously reported in the most proximal portions of highly specialized olfactory sensory cilia (*Williams et al., 2014*). This suggests that cell-type and cilia-specific differences exist. In marked contrast to WT cells, in *Wdr35$^{-/-}$* MEFs, both the myristoylation and palmitoylation (MyrPalm) or dual palmitoylation (PalmPalm) EGFP failed to concentrate in mutant cilia (*Figure 4C*). This failure to recruit lipidated cargos into *Wdr35* mutant cilia is consistent with a more general traffic disruption of ciliary-destined membrane microdomains, containing broad categories of the membrane and membrane-associated cargos.

## WDR35 and other IFT-A complex proteins share close sequence and structural similarity to COPI complex proteins α and ß′

It has previously been suggested that IFTs evolved from a protocoatomer (*Jékely and Arendt, 2006*; *Taschner et al., 2012*; *van Dam et al., 2013*). Three classic coat complexes (COPI, COPII, and clathrin) exist and perform similar functions but on different membranes and follow different routes through the cell. They are made of different protein components, which share a similar division of labor, characterized functionally as either adaptors or cage-forming proteins. Although components like the cage proteins share significant structural homology in organization of protein domains, they do not share detectable sequence homology (*Field et al., 2011*). Given the defects in ciliary membrane content observed in the *Wdr35* mutant cilia, we hypothesized that WDR35, in collaboration with other IFT-A complex proteins, may be required for moving ciliary membrane cargos between donor membranes, such as the Golgi or endosomes, to their destination ciliary membrane, in a manner comparable to coat complexes. WD40 repeat (WDR) and tetratricopeptide repeat (TPR) motifs are common throughout cellular proteomes and are involved in a wide range of biological processes. Agnostic of structure, we used deep sequence analysis of the whole human proteome and homology modeling to ask which proteins were most similar to IFT-A components. Simple alignment strategies with proteins such as IFT subunits, which contain tandem repeat motifs, could erroneously align with other repeat proteins to suggest a close evolutionary relationship where none exists (i.e., false positives). To address this, we used four IFT-A subunits (IFT144, IFT140, IFT122, and IFT121) and two of IFT-B (IFT80 and IFT172) as seed sequences for multiple iterative rounds of homology searches via profile-HMM alignment (*Remmert et al., 2011*). We then clustered the resulting proteins based on sequence similarity, as

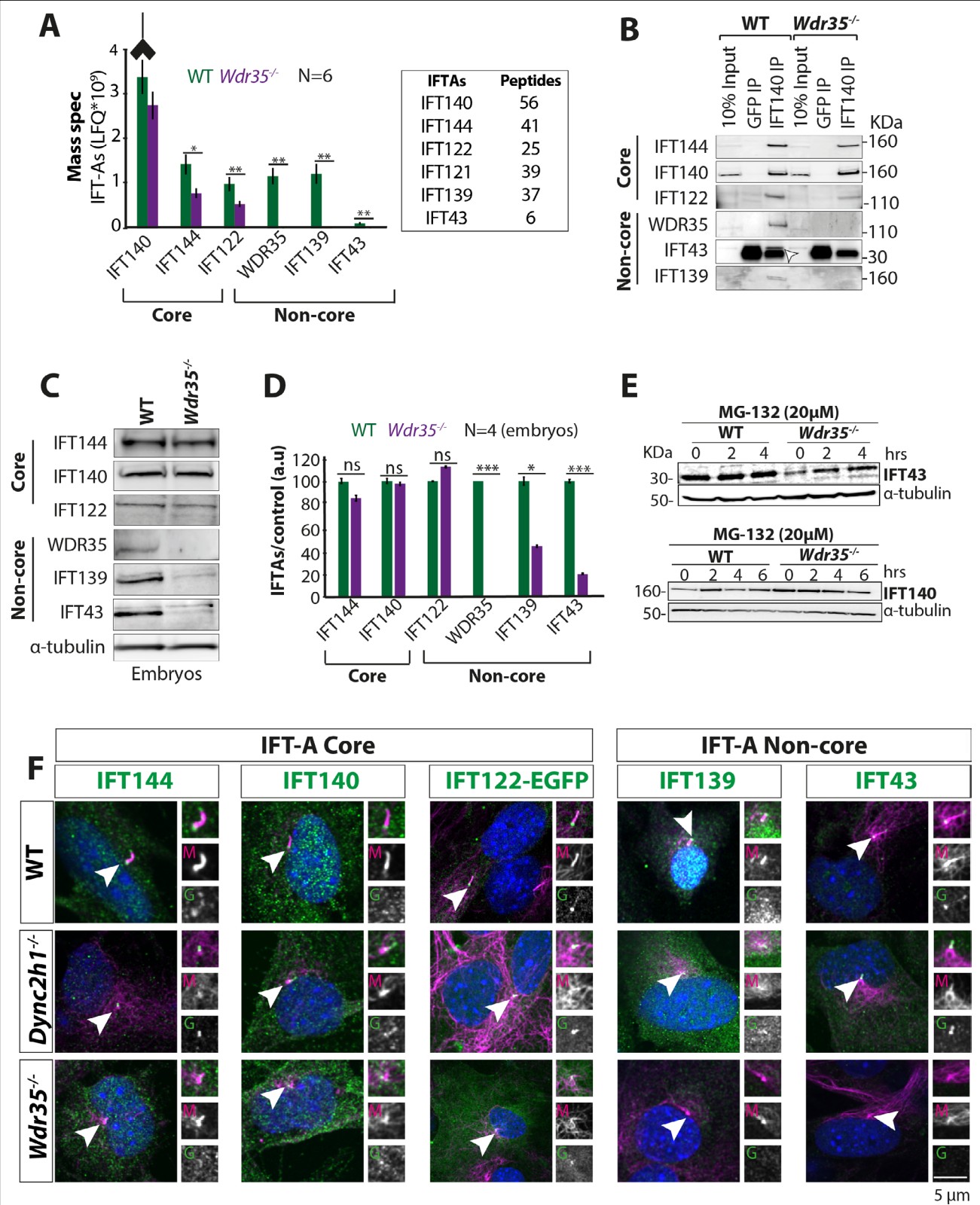

**Figure 3.** WDR35 is essential for the stability and recruitment of the IFT-A complex into cilia. (**A**) Immunoprecipitation/mass spectrometry (IP/MS) data show the stability of the IFT-A complex is disrupted in *Wdr35⁻/⁻* lysates. N = 6 embryos/genotype. Antibody highlights bait (IFT140) for IP. (**B**) Immunoblots confirm the non-core IFT-A complex is unstable in *Wdr35* mutants. IFT43 runs close to the molecular weight of IgG and is shown by an arrow as IFT43 band over the IgG band from IFT140 IP in wild type (WT). The corresponding band is absent in *Wdr35* null samples. (**C, D**) Immunoblots

*Figure 3 continued on next page*

**Figure 3 continued**

for the total level of IFT-A subunits in E11.5 embryo lysates show non-core components IFT139 and IFT43 to be missing in *Wdr35* mutants (**C**), quantified by densitometry (**D**). N = biological replicates. Asterisk denotes significant p-value from t-test: *p<0.05, **p<0.01, ***p<0.001. (**E**) Inhibition of the proteasome by treatment with MG-132 rescues IFT43 stability in *Wdr35$^{-/-}$* mouse embryonic fibroblasts (MEFs). (**F**) MEFs serum-starved for 24 hr reveal a retrograde transport defect in *Dync2h1$^{-/-}$* versus a failed recruitment of IFT-A proteins into *Wdr35$^{-/-}$* cilia. Cells are fixed and stained for respective IFT-A (green) and γ- and acetylated α-tubulin (magenta). Arrowheads point at cilia. Scale bars = 5 µm. Due to a lack of specific immunoreagents, IFT122 signal is from transiently expressed Ift122-GFP. All other panels represent endogenous signal detected by IF.

The online version of this article includes the following source data and figure supplement(s) for figure 3:

**Source data 1.** Full immunoblots labeled and unlabeled for *Figure 3B, C and E*.

**Figure supplement 1.** WDR35 is essential for the stability and recruitment of the IFT-A complex into cilia.

**Figure supplement 1—source data 1.** Full immunoblots labeled and unlabeled for *Figure 3—figure supplement 1A* as well as full data points and stats test of IFT-A and cilia length for *Figure 3—figure supplement 1C*.

previously described (*Wells et al., 2017*; *Wells and Marsh, 2019*). This was repeated using the COP protein subunits as seeds for reverse analysis. Together, these reciprocal analyses revealed that out of the entire proteome COPI α and β' cluster most closely with six IFT proteins (two IFT-B and four IFT-A components), both having TPR and WD40 repeats (*Figure 5A*). In contrast, homology searches with COPI β and COPI ϒ1/2, which have HEAT/ARM repeats, did not yield any hits with IFT components, as was the case with COPI ε, which has TPRs but no WD40 domains. COPI δ and COPI ε1/2, which have no identifiable repeat domains, are most closely related to adaptor protein complex subunits AP2 and AP3. In summary, using multiple rounds of sequence homology searches, we generated a broad set of putatively related repeat proteins, clustering of which reveals clear relationships between coatomers and IFT-A/B complex components.

Next, we used SWISS-MODEL (*Waterhouse et al., 2018*) to predict the structures of IFT-A proteins. COPI α (COPA) and β' (COPB2) structures were top hits with 12–15% sequence identity and 26–27% sequence similarity to four IFT-A complex proteins (IFT144, IFT140, IFT122, and WDR35). Based on the target-template alignment models, built using ProMod3, ribbon diagrams of all four IFT-A subunits modeled structures with two N-terminal seven-bladed WD40 β propellers and C-terminal extended TPRs, also found in α and β' COPI proteins (*Figure 5—figure supplement 1A*), as previously modeled for WDR35 (*Mill et al., 2011*). The remaining two IFT-A subunits were not possible to model accurately. IFT139 contains only TPRs with limited sequence similarity to the ε subunit of COPI coatomer (*van Dam et al., 2013*). IFT43 is the smallest and unstructured protein and could not be modeled and is presumed to be made of α-helices (*Taschner et al., 2012*). While undertaking this work, a crystal structure for IFT80 was published, highlighting that despite the same domain organization IFT80 adopted a distinctive 3D conformation of the second β-propeller domain from β'-COP and also formed a dimer unlike the triskelion COPI cage (*Taschner et al., 2018*). However, purified IFT172 adopted two configurations by negative stain EM when incubated with and without lipids (*Wang et al., 2018*). Thus, respecting the limitations of homology modeling without solved structures, we found four IFT-A proteins (IFT144, IFT140, IFT122, and IFT121) to have very high sequence and structural similarities to COPI α and β' subunits with N-terminal WD40 repeats and C-terminal TPR region (*Figure 5—figure supplement 1A*). Given the structural homology of WDR35 and IFT-As to COPI proteins, which derive vesicles from the Golgi, we asked whether WDR35 and IFT-As were sufficient to directly bind membranes.

## Purified non-core IFT-A can bind directly to specific lipids in vitro

To test if the IFT-A complex directly associates with lipids in vitro, we purified recombinantly expressed IFT-A non-core complex (IFT139/121/43) as well as the dimeric IFT121/43 and the isolated IFT43 subunit of the unicellular organisms *Chlamydomonas reinhardtii* using eukaryotic expression systems (*Figure 5B and C*, *Figure 5—figure supplement 1B–D*). All three samples were soluble, eluted as stoichiometric proteins from size-exclusion chromatography (SEC), and were positively identified by MS. The heterotrimeric IFT-A complex purified from mammalian cells was assessed for lipid binding using membrane lipid strips, detecting any bound protein complex using antibodies against the His-Tag on IFT43. From the protein-lipid overlay results in *Figure 5C*, the His-GFP-tagged IFT-A trimeric complex displays strong binding to phosphatidic acid (PA) as well as weaker binding to phosphatidylserine (PS). Thus the IFT-A trimeric complex binds to these

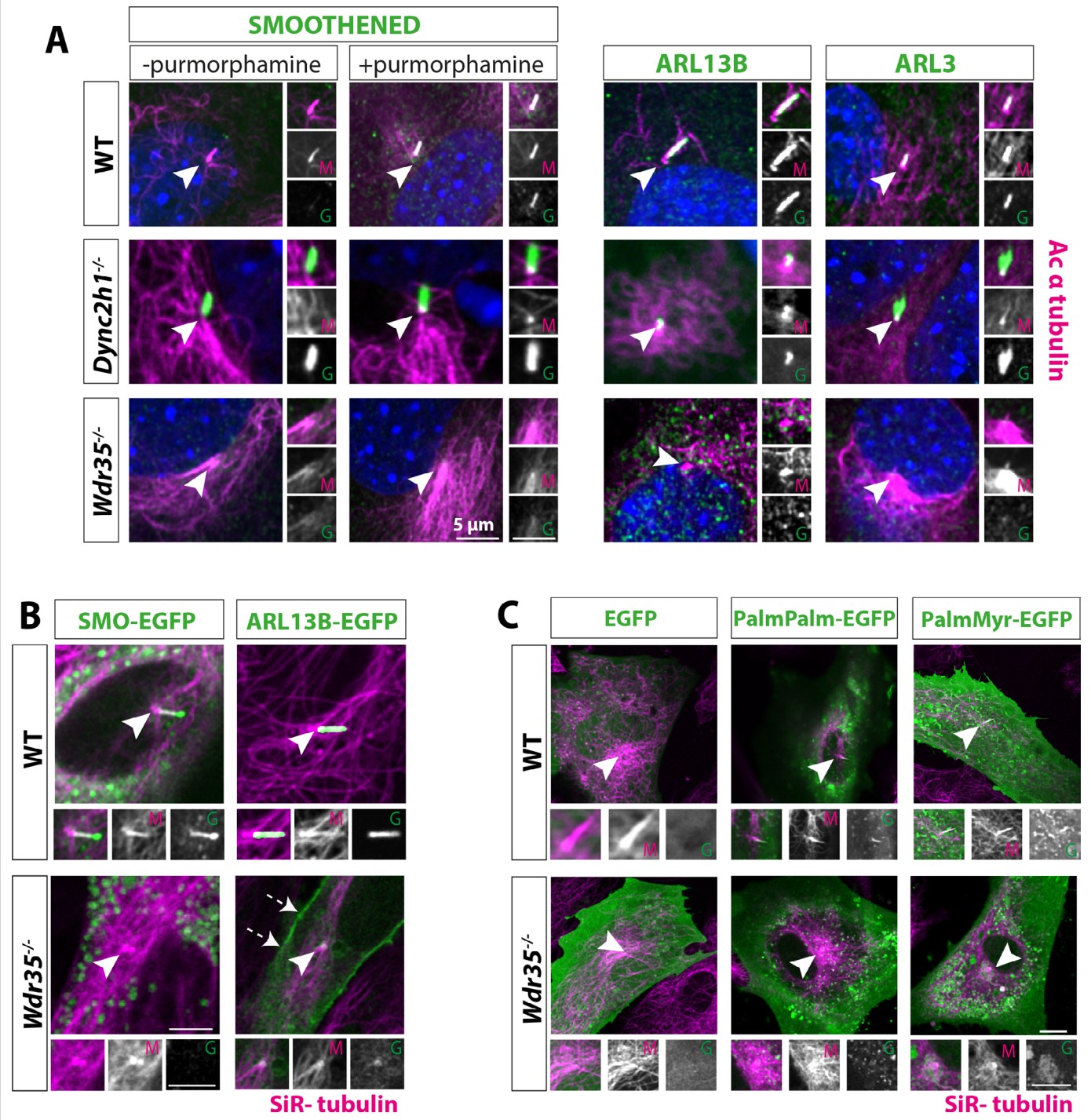

**Figure 4.** Membrane proteins fail to localize to *Wdr35⁻/⁻* cilia. (**A**) 24 hr serum-starved wild type (WT), *Wdr35⁻/⁻*, and *Dync2h1⁻/⁻* mouse embryonic fibroblasts (MEFs) stained for Smoothened (SMO), ARL13B and ARL3 (green), and acetylated α-tubulin (magenta) show failed localization of membrane proteins in *Wdr35⁻/⁻* and retrograde transport defect in *Dync2h1⁻/⁻*. (**B**) Smoothened-EGFP and ARL13B-EGFP (green) expressing ciliated cells stained with SiR-tubulin (magenta) show failed localization of exogenously expressed membrane proteins inside mutant cilia (*Figure 4—video 1*). Dashed arrows point at the enrichment of ARL13B on the membrane in the mutant. (**C**) 24 hr serum-starved cells expressing respective general lipidated GFP cargos (green) and stained for SiR-tubulin show enrichment of lipidated GFP in WT cilia and failed localization in the mutant. Arrowheads point at cilia in all the images. Scale bars = 5 µm.

The online version of this article includes the following video for figure 4:

**Figure 4—video 1.** Cilia-specific membrane-associated cargo (A) ARL13B and membrane-integrated cargo (B) Smoothened (SMO) fail to localize in

*Figure 4 continued on next page*

*Figure 4 continued*

*Wdr35⁻/⁻* cilia.

https://elifesciences.org/articles/69786/figures#fig4video1

negatively charged (anionic) phosphoglycerates exclusively, without binding to neutral or inositol-based lipids as had been reported for the IFT-A adaptor TULP3 (*Mukhopadhyay et al., 2010*). Although there are no reports that PA is a constitutive phospholipid of Golgi apparatus in *C. reinhardtii*, it was shown to be the third most abundant phospholipid in cilia (*Lechtreck et al., 2013*). As a low abundance phospholipid, PA is known to play both important structural roles facilitating membrane curvature during vesicle fusion and fission events (*Arisz and Munnik, 2011*; *Zhukovsky et al., 2019*) as well as signaling functions such as flagellar excision in response to environmental stresses (*Goedhart and Gadella, 2004*; *Lechtreck et al., 2013*; *Quarmby et al., 1992*). To further investigate which subunit of the IFT-A non-core complex is responsible for lipid binding, His-tagged IFT121/43 and IFT43 were also tested in the lipid-strip assay (*Figure 5—figure supplement 1B-D*). Neither the IFT121/43 dimer nor IFT43 alone showed detectable lipid binding, demonstrating that the IFT139 subunit is essential for lipid interaction by the non-core IFT-A complex. In order to further test whether the trimeric IFT-A complex was capable of specifically binding to PA-containing liposomes, we performed negative stain EM of purified proteins incubated with liposomes composed of PE/PG/PA or POPC (control) (*Figure 5D*). The IFT-A trimer was observed to associate with PE/PG/PA liposomes but not to control liposomes (*Figure 5D*). Consistent with the lipid overlay assay (*Figure 5—figure supplement 1B-D*), the IFT121/IFT43 dimer displays only weak association with PE/PG/PA liposomes (*Figure 5D*). The structural homology of IFT-As to COPI proteins and the ability of the non-core IFT-A complex to bind directly to lipids in vitro led us to ask whether IFT-A complex may function similarly to COPI vesicle coats assisting vesicular transport between the Golgi and cilia in vivo.

## Distinct ultrastructural ciliary defects are observed between disruption of IFT-A versus the retrograde IFT motor

We undertook ultrastructural studies to examine traffic phenotypes with higher resolution around cilia in MEFs. In all genotypes, ciliation was observed to start very close to the nucleus and remain close to the Golgi stacks throughout cilia elongation (*Figure 6A*, *Figure 6—video 1*, *Figure 6B*, *Figure 6—video 2*, *Figure 6C*, *Figure 6—video 3*, *Figure 6—figure supplement 2*, *Figure 6—video 4*). In control MEFs, even after 24 hr of serum starvation, very few (~1%) cilia were observed to emerge from the cell, highlighting the deep-seated ciliary pocket in MEFs (*Figure 6B*, *Figure 6—video 2*, *Figure 6—figure supplements 1 and 4A*), and as described for RPE-1 cells (*Molla-Herman et al., 2010*). In control MEFs, polymerized microtubules formed a well-structured axoneme (*Figure 6B*, *Figure 6—video 2*, *Figure 6—figure supplements 1 and 4A*) as previously described in MEFs (*Rogowski et al., 2013*) and reported in other primary cilia (*Kiesel et al., 2020*; *Molla-Herman et al., 2010*). Additionally, microtubules can be seen attached at the cilia base and radiating in different directions in the cell (*Figure 6—figure supplement 1*). In contrast to the well-defined ciliary membrane and well-polymerized microtubules of the control axoneme, *Wdr35⁻/⁻* cilia have 'wavy' membranes and disorganized microtubules (*Figure 6C*, *Figure 6—video 3*, *Figure 6—figure supplement 4B*). Mammalian *Dync2h1⁻/⁻* mutants retained a well-defined ciliary membrane and an apparently well-structured axoneme present throughout (*Figure 6—figure supplement 4C*, *Figure 6—video 5*), similar to previous reports of the *fla14* dynein mutant in *Chlamydomonas* (*Pigino et al., 2009*). Stacked standing trains with a periodicity of 40 nm were reported in *fla-14* mutants (*Pigino et al., 2009*; *Stepanek and Pigino, 2016*), and in our *Dync2h1⁻/⁻* mutant axonemes, we observed similar stacking of stalled IFT trains with a periodicity of 40 nm, irrespective of the length of mutant cilia (*Figure 6—figure supplement 4C*, *Figure 6—video 5*, and *Liem et al., 2012*). Although IFT-Bs also accumulated in *Wdr35⁻/⁻* cilia (*Figure 2A and B*), these stripes were not observed (*Figure 6C*, *Figure 6—video 3*, *Figure 6—figure supplement 4B*), suggesting that both IFT-B and IFT-A are required to form the higher ordered IFT trains that stall in *Dync2h1* mutants.

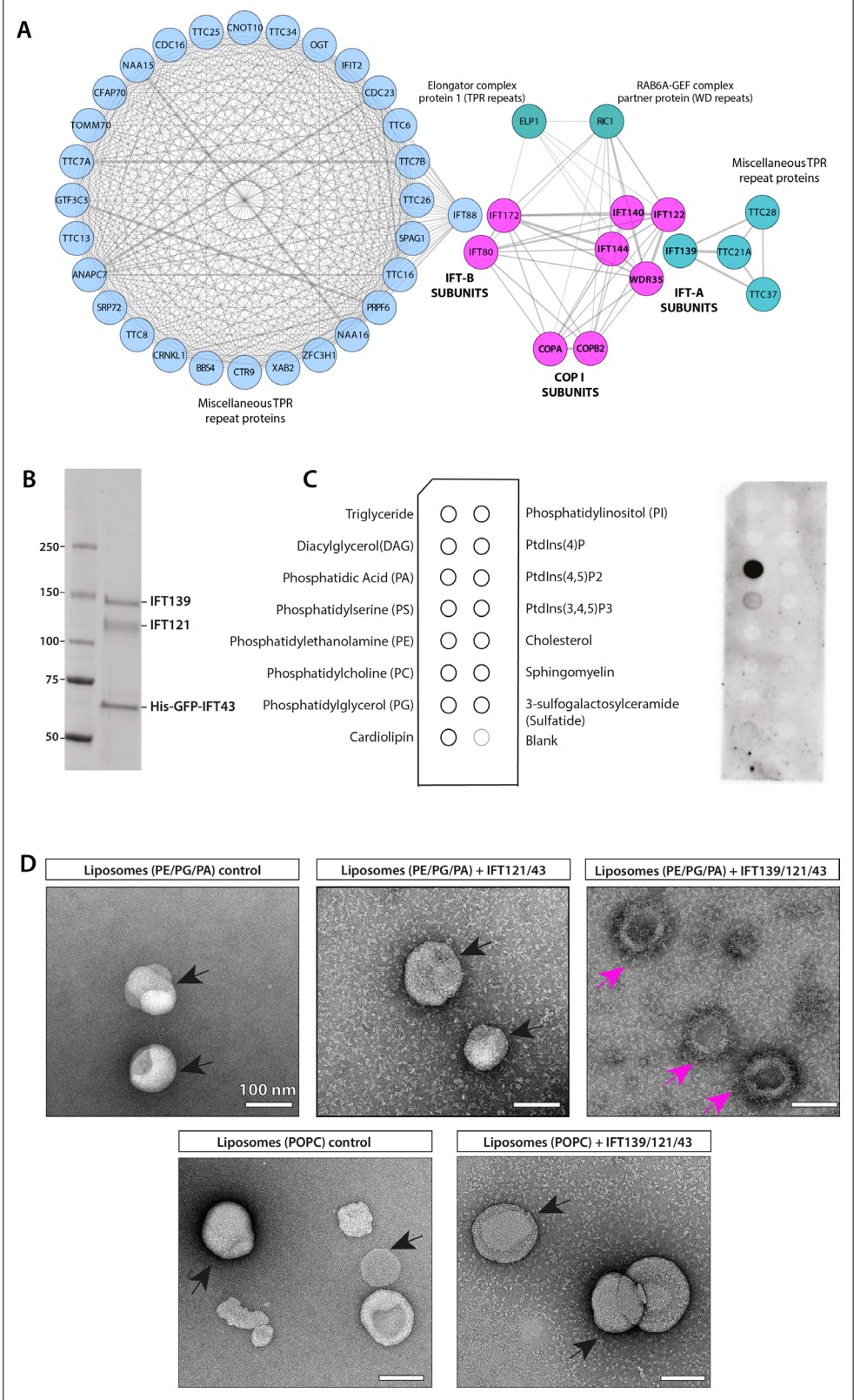

**Figure 5.** IFT-A subunits have close sequence and structural similarity to α and β' COPI subunits and can directly bind to phosphatidic acid (PA) in vitro. (**A**) Clusters of IFT and COPI subunits generated from the results of reciprocal sequence similarity searches with HHBlits using IFT144, IF140, IF122, and WDR35 as initial search queries suggest a very close similarity between a subset of IFT proteins and the COPI α (COPA) and β' (COPB2) subunits.

*Figure 5 continued on next page*

*Figure 5 continued*

Clusters are color-coded according to protein structural motifs with tetratricopeptide repeat (TPR) proteins (blue) and dual WD40 repeat and TPR-containing proteins (magenta). Lines between clusters indicate sequence-based proximity. (**B**) The SDS-PAGE analysis of the purified IFT139/121/His-GFP-43 after purification by size-exclusion chromatography (SEC). (**C**) Lipid-strip overlay assay to detect binding between the IFT-A trimer shown in panel (**A**) and various lipids as indicated in the schematics on the left-hand side of panel (**C**). The IFT-A trimer displays strong binding to PA and weaker binding to phosphatidylserine (PS) in the protein-lipid overlay assay. Both are negatively charged (anionic) phosphoglycerates, whereas the trimer shows no binding to neutral or inositol-based lipids. (**D**) Negative stain micrographs show that the IFT-A trimer (IFT139/121/43) complex associates with liposomes (PE/PG/PA) but not with POPC-liposomes lacking PA. The IFT121/43 dimer associates weakly with liposomes (PE/PG/PA). The particles of liposomes with smooth surfaces are highlighted in black arrows, and liposomes with rough surface displaying protein binding are highlighted in magenta arrows. Scale bar: 100 nm.

The online version of this article includes the following source data and figure supplement(s) for figure 5:

**Source data 1.** Full immunoblots labeled and unlabeled for *Figure 5B and C'* and *Figure 5—figure supplement 1A*.

**Figure supplement 1.** IFT-A subunits have close sequence and structural similarity to α and β′ COPI subunits and can directly bind to phosphatidic acid (PA) in vitro.

**Figure supplement 1—source data 1.** Full immunoblots labelled and unlabelled for *Figure 5—figure supplement 1B-D*.

## WDR35 facilitates formation of coated vesicles containing membrane proteins destined for cilia

We further tested our hypothesis that IFT-A acts as a coat-like complex for vesicles targeted to cilia by transmission electron microscopy (TEM) analysis of ciliated MEFs. We observed electron-dense-coated vesicles between the Golgi and cilia in WT MEFs (*Figure 6A*, *Figure 6—video 1*). We also observed these coated vesicles clustering at the cilia base (*Figure 6B*, *Figure 6—video 2*) and bulging from ciliary pockets and ciliary sheaths in WT MEFs (*Figure 6—figure supplement 1*). These electron-dense vesicles around control cilia were more prominent at the early stage of ciliogenesis in EM (*Figure 6A*, *Figure 6—video 1*).

In contrast, in *Wdr35*[-/-] mutant cells, there is a 10-fold increase in the average number of vesicles around the ciliary base (*Figure 6C*, *Figure 6—video 3*, *Figure 6—figure supplement 2*, *Figure 6—video 4*, *Figure 6—figure supplement 4B*, *Figure 7—figure supplement 1A*; quantified in *Figure 7B*). Importantly, virtually all of these mutant vesicles lack the electron-dense coats observed in control cells (*Figure 6C*, *Figure 6—video 3*, *Figure 6—figure supplement 2*, *Figure 6—video 4*, *Figure 6—figure supplement 4B*, *Figure 7E*, *Figure 7—figure supplement 1B*; quantified in *Figure 7D*). Notably, we did observe other electron-dense coats, likely clathrin, on budding vesicles at the plasma membrane in these same *Wdr35* mutant cells, emphasizing that other coats are preserved in these conditions (*Figure 6—figure supplement 2*, *Figure 6—video 4*, *Figure 7E*). Moreover, no difference in the density or distribution of periciliary clathrin-positive vesicles is observed around the base of *Wdr35*[-/-] mutant cilia (*Figure 7—figure supplement 1C and D*). In contrast, the accumulation of coat-less vesicles spreads in a volume ~2 μm$^3$ around the *Wdr35*[-/-] ciliary base (*Figure 6C*, *Figure 6—video 3*, *Figure 6—figure supplement 2*, *Figure 6—video 4*; quantified in *Figure 7B*, *Figure 7—figure supplement 1A*). In spite of their proximity to the ciliary sheath and their abundance, fusion events were not observed in *Wdr35*[-/-] mutants (*Figure 6C*, *Figure 6—video 3*, *Figure 6—figure supplement 2*, *Figure 6—video 4*; quantified in *Figure 8D*). We do not believe that this periciliary vesicle accumulation phenotype is a general defect in global membrane traffic as the accumulation of vesicles lacking electron densities occurs specifically around mutant cilia, and not at other regions of *Wdr35* mutant cells (*Figure 6—figure supplement 3*, *Figure 6—video 5*).

Clathrin-mediated endocytosis at the ciliary pocket is proposed to regulate internalization of ligand/receptor complexes or membrane content at the base of cilia (*Molla-Herman et al., 2010*). To test whether these vesicles might be important for the import or export of cargo directed to cilia, we analyzed *Dync2h1*[-/-] cilia, which we showed to contain increases in IFTs (*Figures 2 and 3*) and membrane protein cargo (*Figure 4*) in the absence of retrograde transport. Consistent with the redistribution of IFT pools from the base into the ciliary compartment (*Figure 2A* and *Figure 3F*), we observed no vesicles at the base of *Dync2h1*[-/-] cilia (*Figure 6—video 6*, *Figure 7*; quantified in

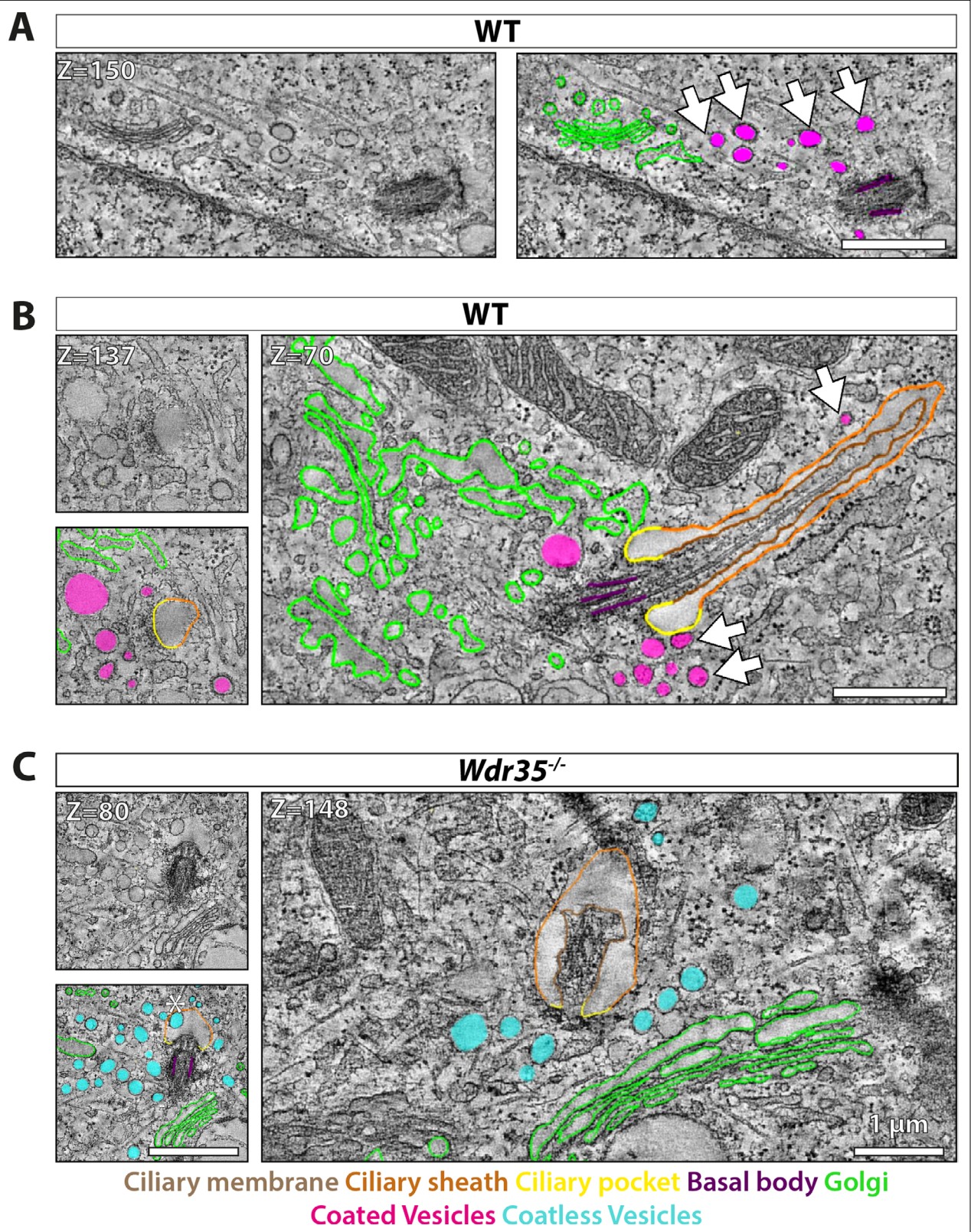

**Ciliary membrane** **Ciliary sheath** **Ciliary pocket** **Basal body** **Golgi**
**Coated Vesicles** **Coatless Vesicles**

**Figure 6.** Electron-dense vesicles are observed tracking between the Golgi and cilia base in wild type (WT) fibroblasts, whereas 'coat-less' vesicles accumulate around *Wdr35* mutant cilia. The tilt series of transmission electron microscopy (TEM) samples were made from 24 hr serum-starved mouse embryonic fibroblasts (MEFs). Reconstructed tomograms are color-coded to highlight the ciliary membrane (brown), ciliary sheath (orange), ciliary pocket (yellow), basal body (purple), Golgi (green), electron-dense-coated vesicles (magenta), and vesicles lacking electron cloud (cyan). (**A**)

*Figure 6 continued on next page*

*Figure 6 continued*

Z-projections from 600 nm TEM serial tomograms of WT MEFs show a track of electron-dense vesicles between the Golgi and cilia (*Figure 6—video 1*). Arrows point at the path of vesicles between the Golgi and cilia. The image in the left panel is segmented in the right panel. (**B**) Z-projections from 300 nm tomograms from WT MEFs show electron-dense-coated vesicles close to the cilia base and along the length of the cilium (*Figure 6—video 2*). Arrows point at coated vesicles near the cilium. (**C**) Z-projections from 600 nm serial tomogram from *Wdr35⁻/⁻* MEFs have a massive accumulation of vesicles in a 2 μm radius of the cilia base (cyan), and these vesicles lack a visible coat, or electron-dense cloud on them (*Figure 6—video 3*). The length of cilia is drastically reduced, the ciliary membrane is wavy, and axoneme microtubules are broken in the mutant. (**B, C**) On left is the same Z-projection in the upper panel segmented in the lower panel, and on the right is another Z-projection from the same tomogram. Asterisk shows a coatless vesicle that fails to fuse with the ciliary sheath (see lower-left panel, **C**). Scale bars = 1 μm.

The online version of this article includes the following video and figure supplement(s) for figure 6:

**Figure supplement 1.** Vesicles with electron-dense coats are observed protruding/fusing with the ciliary sheath in wild type (WT) mouse embryonic fibroblasts (MEFs).

**Figure supplement 2.** Vesicles around cilia in *Wdr35⁻/⁻* mouse embryonic fibroblasts (MEFs) fail to fuse with ciliary pocket or ciliary sheath.

**Figure supplement 3.** Vesicle accumulation/fusion defect around cilia in *Wdr35⁻/⁻* mouse embryonic fibroblasts (MEFs) is observed focally, suggesting that it is not a global membrane traffic defect.

**Figure supplement 4.** Retrograde dynein motor mutant has a different ciliary structure defect than *Wdr35* mutants.

**Figure 6—video 1.** Track of electron-dense vesicles is present between Golgi and cilia in control mouse fibroblast.
https://elifesciences.org/articles/69786/figures#fig6video1

**Figure 6—video 2.** Electron-dense vesicles are observed around the base of cilia in control mouse fibroblasts.
https://elifesciences.org/articles/69786/figures#fig6video2

**Figure 6—video 3.** In *Wdr35⁻/⁻* fibroblasts, an accumulation of small coatless vesicles is present around short cilia.
https://elifesciences.org/articles/69786/figures#fig6video3

**Figure 6—video 4.** In *Wdr35⁻/⁻* fibroblasts, periciliary vesicles fail to fuse with ciliary pocket or ciliary sheath.
https://elifesciences.org/articles/69786/figures#fig6video4

**Figure 6—video 5.** In *Wdr35⁻/⁻* fibroblasts, preciliary vesicles fail to fuse with ciliary pocket or ciliary sheath.
https://elifesciences.org/articles/69786/figures#fig6video5

**Figure 6—video 6.** *Dync2h1⁻/⁻* cilia lack both coated as well as coatless vesicles at the cilia base, whilst ectosomes are seen budding from the tip.
https://elifesciences.org/articles/69786/figures#fig6video6

---

*Figure 8D*). Interestingly, ectosomes, which are previously reported to regulate the content of cilia in a variety of systems (*Cao et al., 2015*; *Nager et al., 2017*; *Wood and Rosenbaum, 2014*), budding from the tip were much more prevalent in *Dync2h1⁻/⁻* cilia than in WT cells (*Figure 6—figure supplement 4C*, *Figure 6—video 6*). We interpret these data as evidence that the coated vesicles around the WT cilia function to transport cargo possibly from the Golgi or via an endosomal intermediate to the cilia. In the absence of WDR35, non-coated vesicles accumulate around the ciliary base, marking a failure in this process in either the formation and/or maintenance of this coat and subsequent fusion at the target ciliary pocket.

To further confirm our hypothesis that these electron-dense vesicles directed to cilia contain WDR35 and IFT-A proteins, we performed correlative light and electron microscopy (CLEM) imaging in *Wdr35⁻/⁻* MEFs expressing WDR35-EmGFP, which we had previously shown to completely rescue cilia phenotypes (*Figure 1A and B*, *Figure 8A*). Expressing WDR35-EmGFP in *Wdr35⁻/⁻* ensures that every WDR35 particle was labeled with EmGFP, minimizing competition with non-labeled species. Using Airyscan confocal imaging of WDR35-EmGFP MEFs grown on grids for subsequent TEM, we saw WDR35-EmGFP enriched at the ciliary base of rescued mutant cilia. Moreover, we observed that this signal coincided with the reappearance of electron-dense vesicles in the TEM images (*Figure 8A and B*). We also observed recovery of fusion events of coated vesicles at the cilia base in cells expressing WDR35-EmGFP as well as rescue of the periciliary vesicle accumulation phenotype (*Figure 8B and C*; quantified in *Figure 8D*, *Figure 8—figure supplement 1*, *Figure 8—video 1*). Next, we performed immunogold labeling directly on 70 nm sections and observed sparse but specific labeling of GFP-positive particles at the cilia base, within the axoneme and around putative vesicles at the cilia base and ciliary sheath (*Figure 8—figure supplements 2 and 3*). Together, these results demonstrate that WDR35 is required for the formation of these coated vesicles and that these coated vesicles coincided with WDR35-EmGFP signal, confirming that WDR35 supports the assembly of a novel coat on vesicles destined to deliver membrane cargos to cilia.

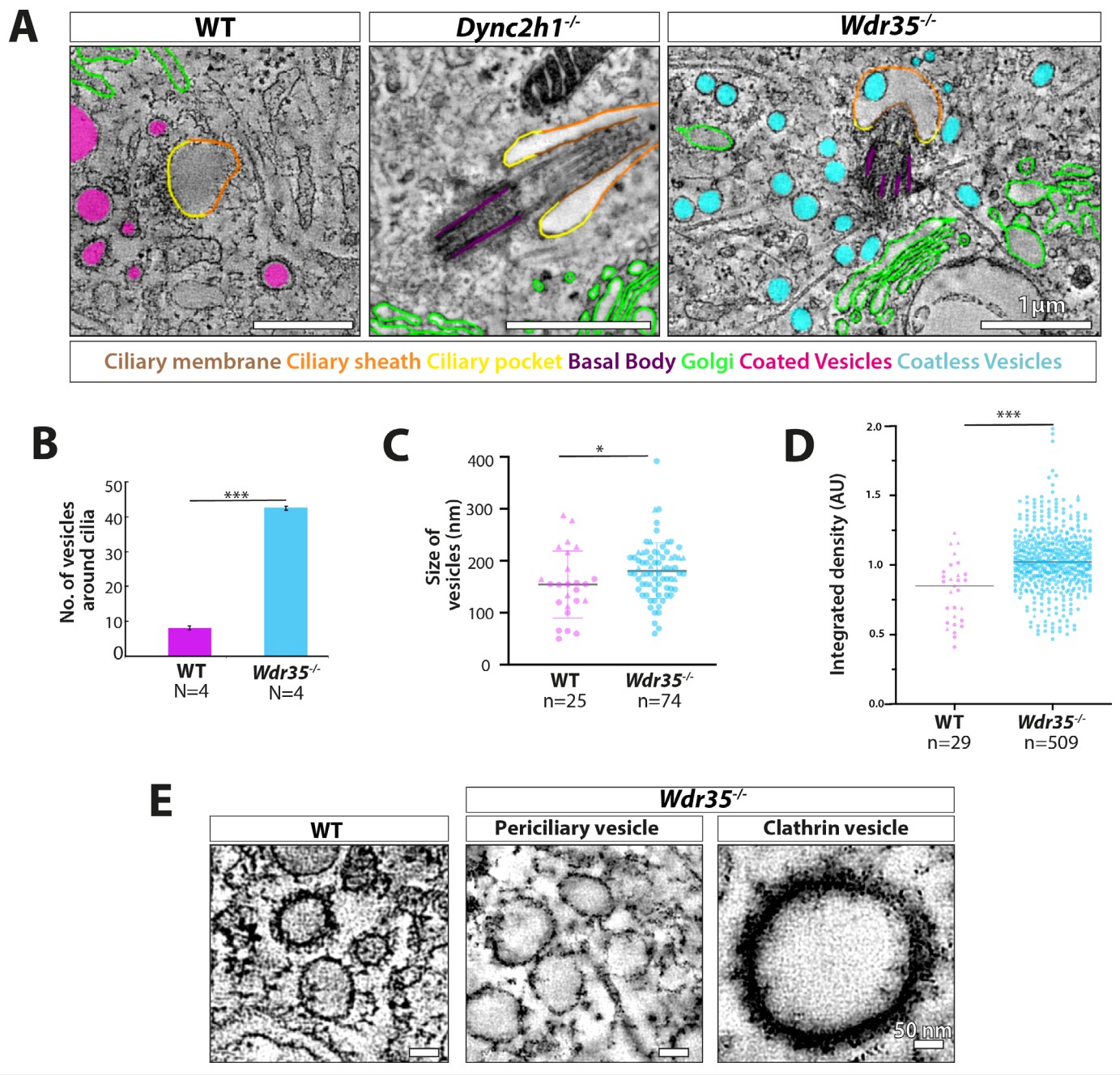

**Figure 7.** Vesicles clustering around *Wdr35*−/− cilia lack electron-dense decorations although electron-dense clathrin-coated vesicles are still observed budding from the mutant plasma membrane. (**A**) Zoomed-in views of periciliary vesicles observed in wild type (WT) (zoomed – **B**, , *Figure 6—video 2*), *Wdr35*−/− (zoomed – **C**, , *Figure 6—video 4*), *Dync2h1*−/− mouse embryonic fibroblasts (MEFs) 24 hr post-serum starvation show vesicles around WT cilia are coated (magenta) and around *Wdr35*−/− are coatless (blue). Very rare vesicles are observed surrounding *Dync2h1*−/− mutant cilia. (**B**) The average number of vesicles around cilia in control and *Wdr35*−/− cells, counted in a volume of 2 µm radius around cilia in transmission electron microscopy (TEM) tomograms, shows 10 times more vesicles in *Wdr35*−/− cells. N = number of whole-cell volume tomograms per genotype. (**C**) The diameter of the periciliary vesicles shows a small but significant increase in size between control and *Wdr35*−/−. n = number of vesicles. The paucity of vesicles around *Dync2h1*−/− cilia prohibited quantification. (**D**) 2D quantification of electron density around vesicles shows signal for control vesicles is lower (darker) than mutant median (lighter) as determined by 20 nm ring outside all annotated objects. (**E**) Zoomed-in images to highlight the difference in the electron-dense cloud surrounding periciliary vesicles in WT (*Figure 6—video 2*), which are largely missing in *Wdr35*−/− (*Figure 6—video 4*, *Figure 6—video 5*) MEFs. Clathrin vesicles from the same mutant (*Figure 6—video 4*) maintain their coat, confirming missing electron density on *Wdr35*−/− periciliary vesicles is not a fixation artifact. Scale bars, (**A**) = 1 µm and (**E**) = 50 nm. N = number of cells examined. n = number of vesicles scored. Asterisk denotes significant p-value from t-test: *p<0.05, **p<0.001, ***p<0.0001.

*Figure 7 continued on next page*

*Figure 7 continued*

The online version of this article includes the following figure supplement(s) for figure 7:

**Figure supplement 1.** Increased periciliary vesicles in *Wdr35* mutant cells are unlikely to be clathrin-based as number and distribution of clathrin-positive foci remain unchanged.

## Discussion

### WDR35 is a component of a novel coat-like complex required for entry of cargos into cilia

Vesicle coat proteins, with the archetypal members clathrin and the coat protein complexes I and II (COPI and COPII, respectively), are macromolecular machines that play two central roles in the homeostasis of the cell's endomembrane system. They enable vesicle formation and select protein and lipid cargo packaged for delivery from a specific donor to functionally segregated compartments. Given the deep sequence structural similarities between IFT-A and COPI subunits and the ability of the non-core IFT-A to bind directly to lipids in vitro, coupled to the phenotypic defects in *Wdr35*[-/-] cells (including lack of ciliary enrichment of a broad range of membrane cargos and the absence of electron-densities on accumulated periciliary vesicles), we propose a novel function for WDR35 and other IFT-A proteins to act as a coat-like complex that is critical for the transport of ciliary membrane cargo into cilia. Two other macromolecular complexes have been proposed to form vesicle-associated coats involved in ciliary traffic: clathrin (*Kaplan et al., 2010*; *Molla-Herman et al., 2010*) and the BBSome complex (*Jin et al., 2010*).

Clathrin is a classical vesicle coating protein with some documented activity at the ciliary pocket (*Clement et al., 2013*; *Pedersen et al., 2016*). From static images, the directionality of events is difficult to resolve: fission (endocytosis) or fusion (exocytosis). Clathrin vesicles can be both endocytic, where they concentrate cargos and curve off donor membranes for selective transport into the cytoplasm, or exocytic, where they can use fuse to release their contents. For example, a subset of AP-1 clathrin vesicles were shown to traffic between the trans Golgi and basolateral membranes of polarized epithelial cells (*Fölsch et al., 1999*) via the recycling endosome compartment (*Futter et al., 1998*). Indeed, in both *C. elegans* (*Bae et al., 2006*; *Dwyer et al., 1998*; *Kaplan et al., 2010*; *Ou et al., 2007*) and trypanosomes (*Vince et al., 2008*), deletion or depletion of AP-1 leads to defects in cilia assembly and protein traffic into cilia. However, in mammalian cells, depletion of clathrin and clathrin-associated proteins results in a normal number of cilia with normal lengths (*Kaplan et al., 2010*; *Molla-Herman et al., 2010*), as opposed to the drastically reduced size of *Wdr35*[-/-] cilia (*Caparrós-Martín et al., 2015*; *Fu et al., 2016*; *Mill et al., 2011*). This suggests that clathrin is dispensable for vesicular transport into mammalian cilia. Although electron-dense vesicles were observed invaginating from the mammalian ciliary pocket, the electron density on these vesicular invaginations was unchanged in the absence of clathrin (*Molla-Herman et al., 2010*). Using live cell imaging, the directionality of clathrin-mediated traffic was reported to be largely away from cilia (*Molla-Herman et al., 2010*). Importantly, we still observe clathrin-coated endocytic structures on the plasma membrane of *Wdr35*[-/-] cells (*Figure 6—figure supplement 2*, *Figure 6—video 4*), and we found no difference in the distribution of clathrin intensity in a volume of ~2 µm³ around the ciliary base in *Wdr35*[-/-] cilia compared to controls (*Figure 7—figure supplement 1C and D*). Moreover, studies on clathrin-mediated exocytosis demonstrated that depletion of human clathrin heavy or light chains results in increased total fusion events with complete release of membrane cargos from vesicles in fibrosarcoma cells (*Jaiswal et al., 2009*), the opposite to what is observed in *Wdr35* mutants where vesicles stack up adjacent to the ciliary sheath but do not fuse.

The BBSome is a macromolecular machine of Bardet–Biedl syndrome (BBS) proteins, which is also postulated to have evolved from an early ancestral coat complex (*Jékely and Arendt, 2006*; *van Dam et al., 2013*). The BBSome shares similar structural elements to the archetypal coats and plays a role in cilia function (*Nachury, 2018*). In contrast to IFT, mutations in BBSome components, including ARL6/BBS3, do not affect cilia assembly and length regulation (*Domire et al., 2011*; *Eguether et al., 2014*; *Lechtreck et al., 2013*; *Lechtreck et al., 2009*; *Liew et al., 2014*; *Nager et al., 2017*; *Shinde et al., 2020*; *Xu et al., 2015*; *Ye et al., 2018*). Instead, they generally are required for regulating cilia content, mostly for the export of ciliary membrane proteins. Although this suggests that BBSomes regulate movement of ciliary components between compartments, endogenous localization of the

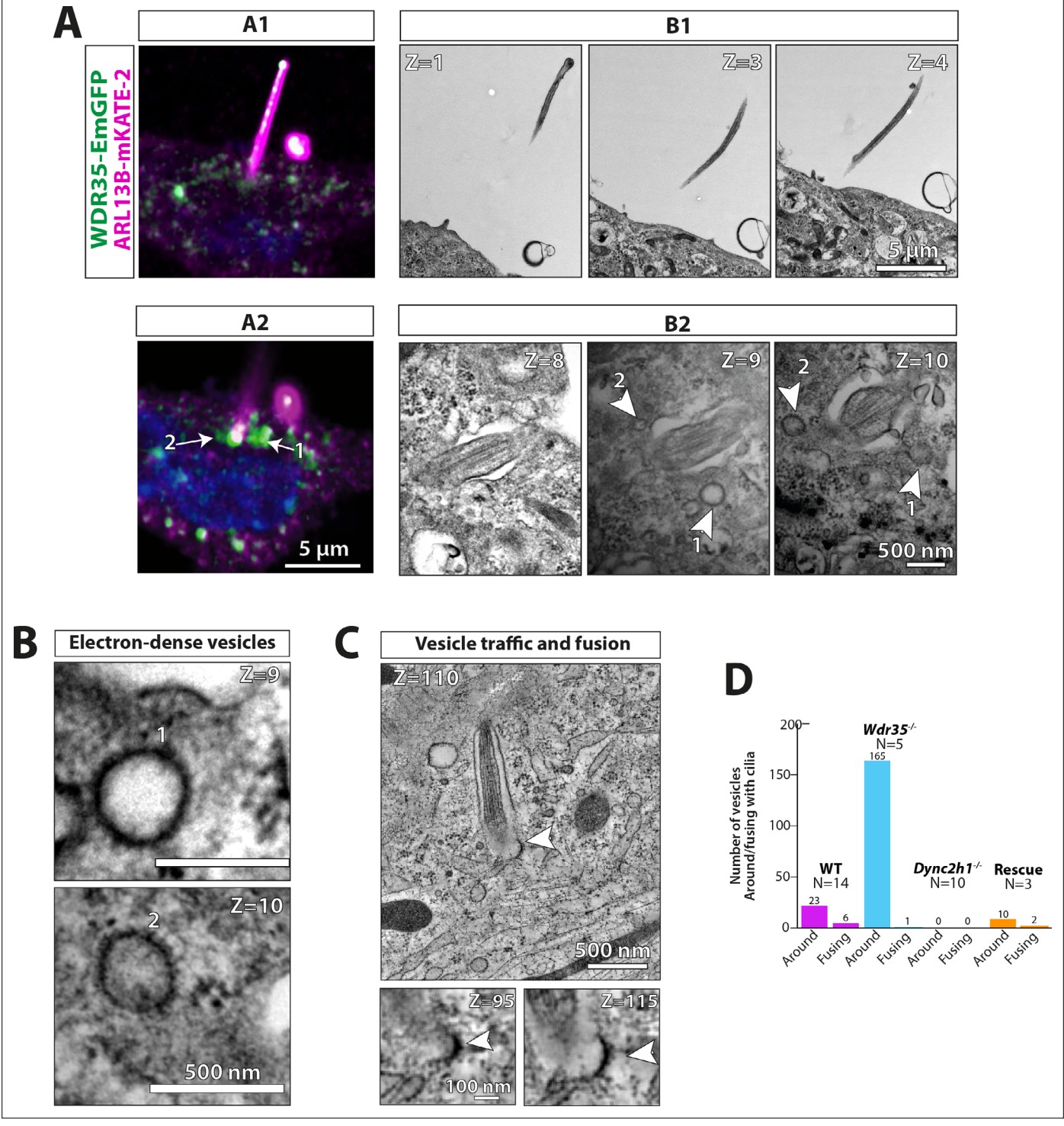

**Figure 8.** WDR35 is sufficient to rescue cilia elongation and restore traffic of coated vesicles, which are GFP-positive by correlative light and electron microscopy. 4 hr serum-starved *Wdr35*⁻/⁻ cells rescued for ciliogenesis by expressing WDR35-EmGFP (green) and imaged first with Airyscan confocal imaging followed by transmission electron microscopy (TEM) imaging. ARL13B-mKATE (magenta) is used as a cilia marker. (**A1**) and (**A2**) represent two sequential Z-stacks from Airyscan confocal imaging. (**B1**) and (**B2**) represent TEM sequential images of 70 nm sections of the same cell. Arrows point at WDR35 localizing close to the cilia base, as shown by LM imaging, whilst arrowheads correspond to electron-dense vesicles shown in Z = 9 and Z = 10 TEM images. (**B**) The same two sections Z = 9 and Z = 10 enlarged in the last panel show two rescued coated vesicles close to cilia. (**C**) Zoomed-out Z-section from 1200-nm-thick TEM tomogram of a different cell expressing Wdr35-EmGFP showing coated vesicle fusing with ciliary pocket (arrowhead) left. Bottom: zoomed-in view of two sections showing electron density on the fusing vesicle (full series shown in *Figure 8—figure supplement 1*,

*Figure 8 continued on next page*

*Figure 8 continued*

*Figure 8—video 1*). (**D**) Quantification of fusion figures observed between genotypes. N = number of cells. See *Figure 8—figure supplement 1*, *Figure 8—video 1*. Scale bars: (**A2**) and (**B1**) are 5 µm, (**B2**) and (**B**) are 500 nm, and (**C**) is 500 nm (upper panel) and 100 nm (lower panel).

The online version of this article includes the following video and figure supplement(s) for figure 8:

**Figure supplement 1.** WDR35 is sufficient to rescue cilia elongation and restore coated vesicles fusion with the ciliary pocket.

**Figure supplement 2.** WDR35 localizes on vesicles around the cilia and concentrates at the ciliary pocket before entering the cilia by immunogold electron microscopy (EM) labeling.

**Figure supplement 3.** WDR35 localization to vesicles around the cilia and ciliary pocket by immunogold electron microscopy (EM) labeling is specific.

**Figure 8—video 1.** WDR35 is sufficient to rescue cilia elongation and restore fusion of coated vesicles.

https://elifesciences.org/articles/69786/figures#fig8video1

BBSome remains unclear, without evidence supporting endomembrane or plasma membrane localization. In contrast, IFT20 localizes to the Golgi (*Follit et al., 2006*; *Noda et al., 2016*). Moreover, whilst there is in vitro evidence that BBSomes can cluster on liposomes, they do not deform membranes, a key step in vesicle formation by coatamers (*Jin et al., 2010*). In contrast, purified IFT172, an IFT-B component that is also homologous to COPI α and ß′ like WDR35, can not only assemble on liposomes with high affinity but can also bud 50 nm vesicles consistent with coatamer-sized products (*Wang et al., 2018*). We report here that the purified trimer of non-core IFT-A (WDR35, IFT43, and IFT139) can also directly and specifically bind to lipids, notably PA, which is involved in membrane deformation in COPI maturation and exocytosis (*Yang et al., 2008*; *Zeniou-Meyer et al., 2007*). We are currently testing whether non-core IFT-A can also pinch off vesicles. Together, the evidence, including its evolutionary conservation of the BBSome with more classical coat proteins (*Jékely and Arendt, 2006*; *van Dam et al., 2013*), interaction with in vitro membranes in the presence of the ARF-like GTPase ARL-6, interaction with phospholipids (*Jin et al., 2010*; *Nachury et al., 2007*), and recent cryo-EM structures of the complex (*Chou et al., 2019*; *Klink et al., 2020*; *Singh et al., 2020*; *Yang et al., 2020*), suggests that the BBSome may be working as an adaptor for IFT-A-mediated cage formation, similar to other coat adaptors for clathrin (i.e., AP1/AP2) or COP (i.e., β-, γ-, δ-, and ζ -COP for COPI). Our data suggest that the electron density observed on vesicles around the ciliary base in control cells is neither clathrin nor BBSome in nature, and is likely composed of WDR35/IFT-A.

## Mechanism of WDR35/IFT-A-assisted vesicle coat function; regulators of vesicular fusion and fission

Our study demonstrates a requirement for IFT-A to deliver ciliary membrane cargo into cilia, potentially by acting as a vesicle coat operating between the Golgi and the ciliary base. Archetypal coatamer protein complexes, including COPII, COPI, and clathrin, concentrate cargo within donor membranes and pinch off vesicles (fission), which then travel to their target organelle membranes, where SNARE and Rab GTPases assist their fusion (*Bonifacino and Glick, 2004*). In these cases, the electron-dense coats are progressively dismantled such that uncoated vesicles can fuse with acceptor membranes, presumably to facilitate access to the fusion machinery, such as SNAREs, on the surface of the vesicle. As a result of interactions with cargo and lipids with the vesicles, there is evidence that the COPI coat can remain stable on membranes after fission. Moreover, this suggests that COPI vesicle uncoating may be incomplete, such that residual COPI on the vesicle surface enables vesicle recognition and tethering necessary for fusion to the correct acceptor membrane (*Orci et al., 1998*). In contrast to the trail of electron-dense vesicles between the Golgi and the base of cilia in control cells, we observed 10 times more vesicles stalled around the cilia base of $Wdr35^{-/-}$ MEFs. These all lack an electron-dense coat, suggesting that these transport vesicles are formed but fail to fuse at the ciliary target membrane in the absence of WDR35.

This raises a question as to why a protein like WDR35, which shares structural homology to fission-inducing proteins, gives phenotypes consistent with a fusion-facilitating protein. One possibility is that while $Wdr35^{-/-}$ MEFs are missing one COPI α/β′- homolog, the other three core IFT-As (IFT144, IFT140, and IFT122) may be sufficient to compensate by providing interaction motifs necessary for the fission of vesicles from donor membranes such as the Golgi. Indeed, we show IFT122 to be upregulated in $Wdr35^{-/-}$ mutant cells, similar to previous reports in *WDR35* patient cells (*Duran et al., 2017*). However, we and others have demonstrated that in the absence of WDR35 the IFT-A complex is

unstable (*Zhu et al., 2017*) such that any core IFT-A coat on the vesicles from donor membranes such as the Golgi may be easily disassembled. It is interesting to note that non-core IFT139 and IFT43 are helical (*Taschner et al., 2012*) similar to SNARE proteins that mediate vesicle fusion with target membranes. Importantly, we show here that these components, which are degraded in the absence of WDR35, could help mediate the fusion of vesicles with the ciliary pocket or base to transfer membrane cargos into the growing cilia sheath. Indeed, we show that purified non-core IFT-A complex is sufficient to specifically bind PA, which is present in ciliary membranes, as well as the Golgi and the recycling endosome compartment (*Farmer et al., 2021*; *Lechtreck et al., 2013*; *Yang et al., 2008*). The lipid composition of membranes is known to determine their curvature (*McMahon and Boucrot, 2015*); PA being conical in shape concentrates on more curved regions of membranes, resulting in nanoscopic-negative curvature such as found in the ciliary pocket (*Zhukovsky et al., 2019*). Moreover, with a small head group, negative charge, and a phosphomonoester group, PA interacts with proteins and lipids in several subcellular compartments that facilitate fission and fusion of membranes (*Zhukovsky et al., 2019*). From our liposome assay, we speculate that IFT139 binding to the IFT121/43 dimer increases the binding affinity to lipids. Indeed, on its own, the IFT-A dimer signal is below the threshold of detection in the protein-lipid overlay assays but observed to weakly associate to PA-containing liposomes. In our purification of the non-core IFT-A complex with the affinity tag on IFT43, only IFT43/121, rather than IFT43/139, was co-purified together with the trimeric complex, indicating that IFT121 interacts with both IFT43 and IFT139, and is responsible for mediating the interactions between IFT43 and IFT139, which is consistent with what has been previously reported (*Behal et al., 2012*; *Zhu et al., 2017*). In the *Wdr35*[-/-] mutant, and likely IFT139 or IFT43 KO strains, the non-core IFT-A complex will not form, which leads to non-coated vesicles (*Figure 9*). Important next steps will be to systematically investigate vesicular traffic defects in other IFT-A mutants, as well as identify the GTPase that acts to drive formation, uncoating, and fusion of these vesicles.

Recruitment, remodeling, and regulation of protein coats involve cycles of GTP hydrolysis, for example, ARF1 regulates recruitment to membranes of the COPI coatomer (*Dodonova et al., 2017*). It is interesting to note that we and others have been unable to purify IFT-A complex with any GTPases (*Mukhopadhyay et al., 2010*), suggesting that any interaction is transient. This is even in conditions where we can purify endogenous IFT-B complexes with its associated GTPases IFT22/RABL5 and IFT27/RABL4. In COPI, recruitment of coat components to donor membranes starts with the insertion of small GTPase ARF1 into membranes (*Dodonova et al., 2017*). So far only one ARF, ARF4 acting at the TGN (*Mazelova et al., 2009*; *Wang et al., 2017*), has been implicated in ciliary traffic. However, it plays non-ciliary roles and shows early lethality in mouse knockouts without affecting cilia assembly (*Follit et al., 2014*). Mutations in several related ARLs have defects in cilia structure and/or content, including ARL3, ARL6, and ARL13B (*Alkanderi et al., 2018*; *Cantagrel et al., 2008*; *Fan et al., 2004*). At least in the case of ARL13B and ARL3, they fail to accumulate and/or enter mutant cilia, even when overexpressed in the absence of WDR35, although periciliary vesicular staining can be observed. Rab GTPases have been implicated in the ciliary targeting of vesicular cargos (*Blacque et al., 2018*). Notably, expression of dominant negative RAB8 in *Xenopus* photoreceptors (*Moritz et al., 2001*) results in a strikingly similar accumulation of vesicles to our *Wdr35* mutants, which fail to fuse with the ciliary base. Similarly, in RPE-1 cells, dominant negative RAB8 impairs traffic of ciliary membrane cargos (*Nachury et al., 2007*). However, functional redundancy between RABs may exist as neither single nor *Rab8a;Rab8b* double mutant mice have defects in cilia formation. On the other hand, defects in ciliation were observed when *Rab10* was additionally knocked down in *Rab8a;Rab8b* double mutant cells (*Sato et al., 2014*). Excitingly, our work demonstrates IFT-As to be important for the later stage of ciliogenesis, similar to GTPases like RAB23 (*Gerondopoulos et al., 2019*) or RSG-1 (*Agbu et al., 2018*; *Toriyama et al., 2016*). Given that these GTPases have also been shown to sequentially interact with CPLANE subunits INTU and FUZ, which are also required for IFT-A holo-complex assembly (*Gerondopoulos et al., 2019*; *Toriyama et al., 2016*), they will be priorities for future investigations.

We have demonstrated that an IFT-A-dependent coat for membrane vesicles exists and is necessary for their fusion with the ciliary sheath and ciliary pocket, which is continuous with the ciliary membrane. We also showed that this coat is necessary to efficiently deliver cilia-destined signaling molecules into the elongating axoneme of the cilium. This raises the possibility that some of this IFT-A-dependent coat may remain upon vesicle fusion as a now linear 'train' carrying membrane cargos

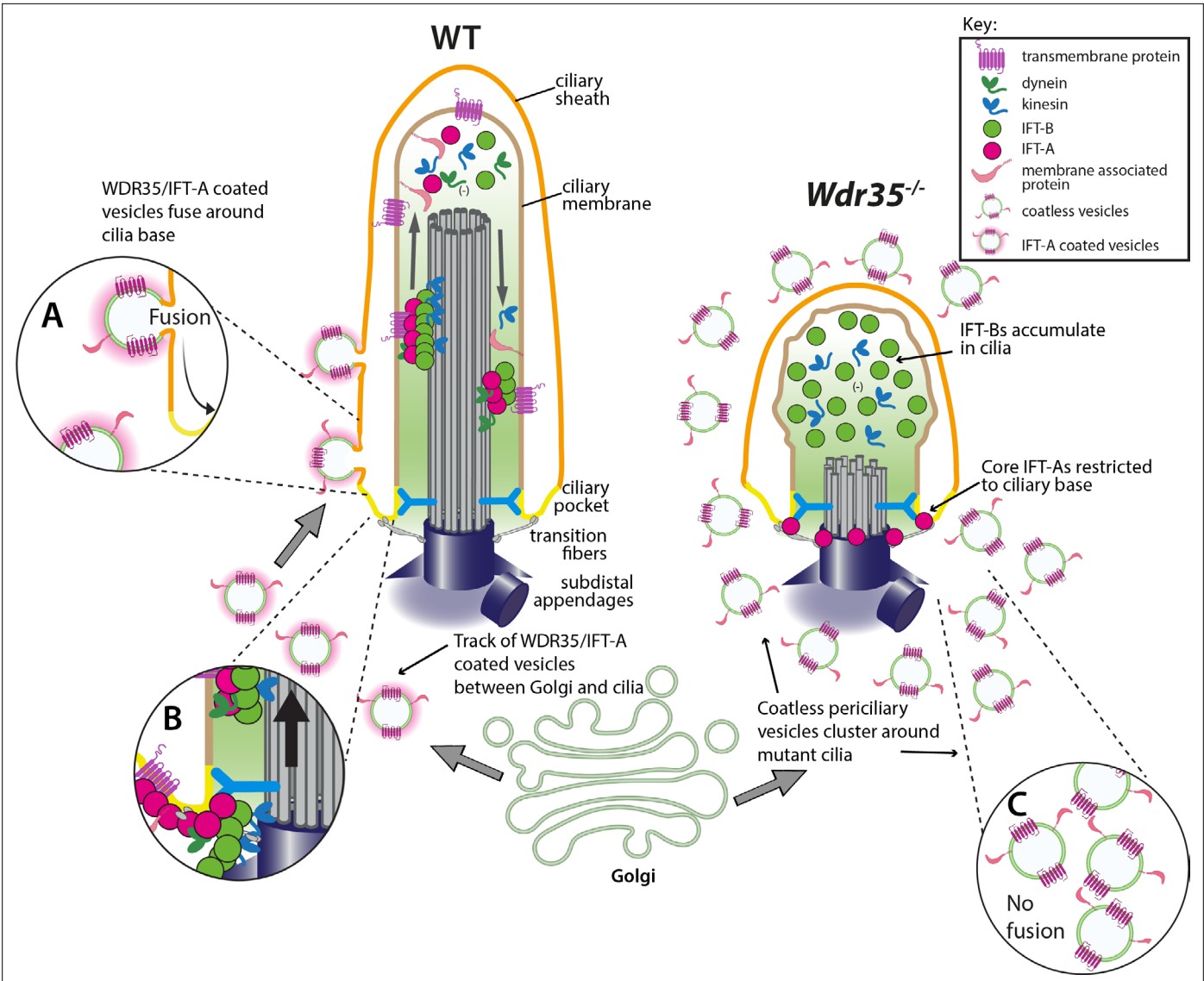

**Figure 9.** WDR35 and likely other IFT-As assist cargo transport of vesicles between the Golgi into cilia at the stage of cilia elongation. Diagrammatic representation of the transmission electron microscopy (TEM) data showing vesicles (green) with the WDR35-dependent coat (magenta halo) fusing and localizing around cilia in wild type (WT) cells (inset A) and coatless vesicles clustering around cilia in *Wdr35*$^{-/-}$ mouse embryonic fibroblasts (MEFs) (inset C). Vesicles follow a track between the Golgi and ciliary base in the WT cells but accumulate without fusing around cilia in *Wdr35*$^{-/-}$ cells. Upon fusion, any remnant IFT-A-dependent coat would become a linear 'train,' which could assemble with cytosolic motors and IFT-B particles for ciliary import across the transition zone (inset B). Without non-core IFT-As, IFT-A core components are restricted at the base of *Wdr35*$^{-/-}$ cilia whilst IFT-B proteins accumulate in short mutant cilia, without any enrichment of ciliary membrane proteins indicating an arrest at the later stages of ciliogenesis during cilia elongation.

to be picked up by cytosolically assembling IFT-B particles allowing import across the transition zone and then anterograde IFT within the cilium (*Figure 9*, inset B). Excitingly, we show that purified non-core IFT-A including WDR35 is sufficient to bind selectively to PA. This low abundance signaling lipid has well-described roles in vesicle traffic where it promotes COPI vesicle fission in the Golgi (*Yang et al., 2008*), maintenance of the endosome recycling compartment (ERC) (*Farmer et al., 2021*), as well as promoting exocytosis through formation of fusion-competent granules (*Zeniou-Meyer et al., 2007*). Defining at which points in vesicular traffic IFT-A-dependent coats act, both fission and fusion, within cells as well as the biochemical nature of lipids and cargos these vesicles carry will be required. Given its efficacy, this IFT-dependent 'targeted delivery' module may also be repurposed for other non-ciliary membrane targeting events via polarized exocytosis. Notably in the immune synapse of T

cells, where IFT20 is required for rapid clustering of TCRs necessary for T cell activation (*Finetti et al., 2009*), as well as photoreceptor dendrites in which IFT localization to vesicles tracking towards the postsynaptic membranes was observed (*Sedmak and Wolfrum, 2010*), where dendritic exocytosis is implicated in synaptic plasticity and neuronal morphology (*Kennedy and Ehlers, 2011*). Future studies into this IFT-dependent coat complex and the membrane traffic processes it controls may expand our phenotypic understanding of the ciliopathies beyond the cilium.

## Materials and methods
### Preparation of primary MEFs, cell culture, ciliation, and genotyping
Primary MEFs were harvested from E11.5 embryos and cultured in complete media (Opti MEM-I [Gibco, 31985-047] supplemented with 10% fetal calf serum (FCS) and 1% penicillin-streptomycin [P/S] and 0.026 µl β-mercaptoethanol) and incubated at 37°C in a hypoxic incubator (3% $O_2$ and 5% $CO_2$). To induce ciliogenesis, 70–80% confluent cells were serum-starved for 24 hr. Genotyping was done as described before for the *Wdr35* line (*Mill et al., 2011*) and *Dync2h1* line (*Caparrós-Martín et al., 2015*). *Pcm1-SNAP* mouse line was made by Dr. Emma Hall (Hall E. et al., unpublished) by endogenous tagging of PCM1 by CRISPR. *Pcm1$^{SNAP}$* mouse line was crossed with *Wdr35$^{-/+}$* and genotyped to screen E11.5 embryos homozygous for both *Wdr35$^{-/-}$* and *Pcm1$^{SNAP/SNAP}$*. MEFs prepared from these embryos were used to image PCM1 localization in WT and *Wdr35$^{-/-}$* using antibodies and other reagents listed in *Table 1* and *Table 2*.

### Electroporation of MEFs
Cells were trypsinized to a single-cell suspension and resuspended in 10 µl Resuspension Buffer R per $0.5 \times 10^5$ cells/transfection reaction, mixed with plasmid DNA (0.75 µg/transfection) (*Table 3*) and electroporated (voltage 1350 V, width 30 ms, one pulse) using a Neon Nucleofection kit (Thermo Fisher Scientific MPK-1096), according to the manufacturer's protocol. Transfected cells are harvested or visualized 24–48 hr post electroporation.

### Live-cell imaging
Primary MEFs ($0.5 \times 10^5$ cells/transfection) were electroporated with ARL13B-EGFP or Smoothened-GFP using the Neon Transfection System, 10 µl kit (Thermo Fisher Scientific, MPK-1096) and seeded in 24-well glass-bottomed plates (Greiner Sensoplates, 662892) with prewarmed media (Opti-MEMI [1×] [Gibco, 31985-047], 10% FCS and 0.026 µl β-mercaptoethanol). Samples were incubated in antibiotic-free media 37°C/5% $CO_2$/3% $O_2$ overnight and then serum-starved for 24 hr. SiR-tubulin kit (Spirochrome, SC002), a 1 mM stock solution, was prepared in anhydrous DMSO and stored at –20°C, without aliquoting. For staining, 1:5000 (200 nmol) of SiR-tubulin stock was diluted in serum-free media and added to cells for 1 hr in the hypoxic incubator, then live imaged without washing.

For live-cell PCM1 imaging, MEFs electroporated with ARL13B-EGFP were incubated with 1:1500 TMR-SNAP (New England Biolabs, S9105S, stock 30 nmol) in low serum media in the hypoxic incubator for 30 min. Cells were washed twice with low serum media for 1 hr each in the incubator. Samples were then incubated for 1 hr in 1:5000 SiR-tubulin (200 nmol). Hoechst 344442 (Thermo Fisher Scientific, H1399) was added 10 min before imaging. Plates were allowed to equilibrate in the Okolabs stage top incubator before confocal imaging on the Leica SP5 using the LAS-AF software, 405 nm diode, argon and 561 and 648 nm laser lines, three Photomultiplier tubes, and one HyD GaSP detector, as per the requirement of the experiment. Images were scanned using a 63× 1.4 NA oil immersion objective and processed using ImageJ and Imaris software.

### Endogenous IFT IPs
Embryos were lysed and homogenized in IP lysis buffer (10 µl/mg) at 4°C on a rotator for 30 min. Composition of IP lysis buffer is (50 mM Tris-HCl [pH 7.5]), 100 mM NaCl, 10% glycerol, 0.5 mM EDTA, 0.5% IGEPAL, and 1/100 Halt protease and phosphatase inhibitor (Thermo Fisher Scientific, 78443) and a tablet of Protease Inhibitor Tablet – one tablet per 10 ml (cOmplete Mini, Roche, 11836170001). The lysate was cleared by spinning at 4°C, 14,000 rpm, for 20 min. The protein concentration was determined using the BCA Protein Assay Kit as per the manufacturer's instruction (Thermo Fisher Scientific, 23225). For each IP, 500 µg of protein was incubated with 3 µg of each antibody overnight at

eLife Research article

Cell Biology

**Table 1.** List of primary antibodies.

| Antibody | Species | Source | Dilution | Fixation |
|---|---|---|---|---|
| ARL3 | Rabbit | Proteintech 10961-1-AP | 1:100 IF | PFA |
| ARL13B | Rabbit | Proteintech 17711-1-AP | 1:100 IF | PFA |
| Ac α-tubulin | Mouse | Sigma T6793 | 1:1000 IF | PFA/GA |
| Ac α-tubulin | Rabbit | Abcam ab179484 | 1:1000 IF | PFA/GA |
| γ-Tubulin | Mouse | Sigma T6557 | 1:100 IF | PFA |
| γ-Tubulin | Rabbit | Abcam ab11317 | 1:100 IF | PFA |
| GFP | Mouse | Roche 11814460001 | 1:1000 IF 1:1000 WB | PFA |
| GFP | Rabbit | Abcam ab6556 | 1:1000 IF 1:1000 WB 1:20 ImEM | PFA |
| GFP | Rabbit | Santa Cruz sc-8334 Stock 0.2 µg/µl | 1:5000 WB 3 µg IP | - |
| His-tag | Mouse | GenScript A00186 | 1:2500 WB | - |
| IFT43 | Rabbit | From Victor L. Ruiz-Perez, University of Madrid, Spain | 1:200 IF 1:2000 WB | PFA |
| IFT81 | Rabbit | Proteintech 11744-1-AP | 1:200 IF 1:1000 WB | PFA |
| IFT88 | Rabbit | Proteintech 13967-1AP Stock 0.23 µg/µl | 1:200 IF 1:1000 WB 3 µg IP | PFA |
| IFT121 | Rabbit | Custom made from Proteintech | 1:50 WB Not for IF | - |
| IFT122 | Rabbit | Proteintech 19304-1-AP | 1:200 IF 1:1000 WB | PFA |
| IFT122 | Rabbit | Aviva ARP 53817_P050 | Not for IF 1:1000 WB | - |
| IFT139 | Rabbit | Novus-NBP1-90416 | 1:1000 WB 1:200 IF | PFA |
| IFT140 | Rabbit | Proteintech 17460-1-AP | 1:200 IF 1:1000 WB 3 µg IP | PFA |
| IFT144 | Rabbit | Proteintech 13647-1-AP | 1:200 IF 1:1000 WB | PFA |
| MKS1 | Rabbit | Proteintech 16206-1-AP | 1:100 IF | MeOH |
| NPHP1 | Mouse | 640 from Greg Pazour, University of Massachusetts | 1:100 IF | MeOH |
| PCM1 | Rabbit | Proteintech 19856-1-AP | 1:100 IF | MeOH |
| Rootletin | Goat | Santa Cruz sc-67828 | 1:100 | MeOH |
| SNAP | Rabbit | New England Biolabs P9310S | 1:300 IF | MeOH |
| α tubulin | Rabbit | Abcam ab4074 | 1:5000WB | |

**Other reagents**

| | | | | |
|---|---|---|---|---|
| SiR-tubulin | - | Spirochrome (SC002) | 200 nM | Live-cell imaging |
| SNAP-TMR | - | New England Biolabs SNAP-Cell TMR-STAR (S9105S) | 1 µM | Live-cell imaging |

*Table 1 continued on next page*

PFA, paraformaldehyde; GA, glutaraldehyde; ImEM, immuno electron microscopy; IF, immunofluorescence; WB, western blot; IP, immunoprecipitation; MeOH, methanol.

4°C (Table 1) with mild agitation (side-to-side). IP of immunocomplexes was done using PureProteome Protein G magnetic beads (Millipore LSKMAGG10). 30 μl beads/IP were equilibrated with 500 μl IP lysis buffer by gentle agitation for 5 min at 4°C. Tubes were placed on a magnet for 2 min, and the buffer was aspirated off with the fine pipette. 200 μl antibody-lysate mix was added to each tube of 30 μl equilibrated beads and incubated for 45 min with agitation to concentrate immunoglobulin complexes on beads at 4°C. Washes (eight times) were performed, each lasting 5 min. Washes were as follows: 2× washes in Wash Buffer-1 (same as IP lysis buffer), followed by 2× washes with Wash Buffer-2 (IP lysis buffer with reduced 0.2% IGEPAL), finally 4× washes with Wash Buffer-3 (IP lysis buffer without any IGEPAL detergent). All wash buffers were aspirated, and dry beads were stored at –80°C, or samples were sent immediately for MS.

## Mass spectrometry

All MS experiments were done at the IGMM Mass Spectrometry facility as per their published protocol (Turriziani et al., 2014). Briefly, the immunocomplexes collected on magnetic beads were processed to generate tryptic peptides. Proteins were eluted from beads by incubating at 27°C for 30 min in elution buffer (2 M urea, 50 mM Tris-HCl pH 7.5, and 5 μg/ml trypsin). The sample was centrifuged, bead pellets washed twice, and the supernatant from samples digested overnight at room temperature (RT). Iodoacetamide was added to the samples to inhibit disulfide bond formation and incubated for 30 min in the dark. Following this, trifluoroacetic acid (TFA) was added to stop tryptic digestion. Desalting and pre-fractionation of the digested peptides were done by manually using C18 pipette stage-tips filled with 3 M Empore disc activated with 50% acetonitrile and 0.1% TFA and then washed once with 0.1% TFA. The peptide mixtures were passed manually along to the column with a syringe to concentrate and purify the analytes. Peptides were subsequently eluted twice in 50% acetonitrile and 0.1% TFA and both eluates were combined. Samples were concentrated and resuspended in 0.1% TFA. This was followed by chromatographic separation on a Reprosil column along a 3–32% acetonitrile gradient. The LC setup was attached to a Q-Exactive mass spectrometer, and ion mass spectra were obtained following HPLC during a tandem MS run. Mass spectra were analyzed using MaxQuant software. Label-free quantification intensity (LFQ) values were obtained for analysis by identifying mass/charge ratio, and their intensities at a specific elution time for individual peptides. The data were collected for both control (GFP) and specific proteins IPs (i.e., IFT88, IFT140; Table 1). LFQ values for the proteins were obtained by summing the ion intensities corresponding to peptides after assigning the unique peptides to proteins. The ratio of LFQ intensities of test:control was taken, where higher the ratio better corresponds to a better enrichment of protein in complex. Complete MS data are available on ProteomeXchange (PXD022652). The relative concentration of IFTs was calculated after normalizing the individual test values with respective GFP-LFQs, as shown in the figures.

## Western blots

Cells or tissues were lysed in 1× Cell Lysis Buffer with the addition of 1/100 Halt protease and phosphatase inhibitor (Thermo Fisher Scientific, 78443) and a cOmplete Protease Inhibitor Tablet, one tablet per 10 ml (cOmplete Mini, Roche, 11836170001). Prepare 1× Cell Lysis Buffer by

**Table 3.** List of plasmids.

| Plasmid | Source |
|---------|--------|
| Arl13b-mKate2 | Diggle et al., 2014 |
| ARL13B-EGFP | Hori et al., 2008 |
| pEGFP-N1 | Clontech 6085-1 |
| Ift122-EGFP | Qin et al., 2011 |
| PalmPalm-EGFP | Williams et al., 2014 |
| MyrPalm-EGFP | Williams et al., 2014 |
| pEGFP-mSmo | Chen et al., 2002 |
| Wdr35-EmGFP | Mill et al., 2011 |

**Table 2.** List of secondary antibodies.

| Antibody | Host | Source | Dilution | Application |
|---|---|---|---|---|
| ECL α-mouse IgG, HRP-conjugated | Sheep | GE Healthcare NA931-1ML | 1:10,000 | WB |
| ECL α-mouse IgG, HRP-conjugated | Rabbit | Dako P0260 | 1:1000 | SO |
| ECL α-rabbit IgG, HRP-conjugated | Goat | GE Healthcare RPN4301 | 1:10,000 | WB |
| α-rabbit light-chain-specific HRP-conjugated | Mouse | Millipore MAB201P | 1:10,000 | WB |
| α-rabbit IgG light-chain-specific mAb | Mouse | Cell Signaling Technology L57A3 | 1:10,000 | WB |
| Alexa 488, 594, 647 conjugated-α-mouse | Donkey | Molecular Probes | 1:500 | IF |
| Alexa 488, 594, 647 conjugated-α-rabbit | Donkey | Molecular Probes | 1:500 | IF |
| Alexa 488, 594, 647 conjugated-α-goat 10 nm gold conjugated-α-rabbit (batch 008721) | Donkey Goat | Molecular Probes BBI Solutions EM GAR10/0.25 | 1:500 1:30 | IF ImEM |

WB, western blot; IF, immunofluorescence; ImEM, immuno electron microscopy.

diluting 10× stock in ddH$_2$0 (Cell Signaling Technology [10x #9803]: 20 mM Tris-HCl [pH 7.5], 150 mM NaCl, 1 mM Na$_2$ EDTA, 1 mM EGTA, 1% Triton-X100, 2.5 mM sodium pyrophosphate, 1 mM β-glycerophosphate, 1 mM Na$_3$VO$_4$, 1 μg/ml leupeptin). The lysate from embryos was homogenized at 4°C for 30 min and from cells was sonicated briefly (5×, 10 s pulses, Bioruptor Diagenode) to lyse the tissue or cells. The lysate was centrifuged at 14,000 g at 4°C for 30 min and the supernatant transferred to a fresh tube. Ready-to-use SDS-PAGE gels (NuPage Novex precast gels, Thermo Fisher Scientific) were used to separate proteins.

The resolved proteins on the gel were transferred to PVDF (Hybond P, GE HealthCare) using the XCell II Blot module as per the manufacturer's instruction. The membrane was then blocked with a 10% solution of dried skimmed milk (Marvel Premier Foods) made in 1× TBST (0.05% Tween-20 in TBS) for 1 hr RT, washed with PBS and incubated with primary antibody (*Table 1*) diluted in 1% skimmed milk solution in 1× TBST overnight at 4°C on shaker/roller. Membranes were washed in 1× TBST 3, 10 min followed by a 1× wash with PBS, and incubated in HRP-conjugated secondary antibody from appropriate species (*Table 2*) for 1 hr at RT, diluted in a solution of 1× TBST and 1% milk. Blot was then washed with 1× TBST, three times and with PBS twice. After the washes, signals were detected by the Super Signal ELISA Femto kit (Thermo Fisher Scientific, 37074) or Super Signal ELISA Pico kit (Thermo Fisher Scientific, 37069). Protein bands were visualized digitally by transmission light imaging on ImageQuant LAS 4000 (GE HealthCare) and analyzed using ImageQuant TL software. Protein bands on blots were quantified with ImageJ/Fiji software by measuring individual bands intensity and normalizing intensities with loading control bands on the same blot.

## Immunofluorescence

Cells were washed two times with warm PBS, then fixed in either 4% PFA in 1× PHEM/PBS 15 min at RT, 2% fresh glutaraldehyde in 1× PHEM for 15 min, or pre-extracted for 30 s on ice in PEM (0.1 M PIPES pH 6.8, 2 mM EGTA, 1 mM MgSO$_4$) prior to fixing in ice-cold methanol on ice for 10 min according to *Table 1*, then washed twice with PBS. 1× PHEM (pH 6.9) contains 60 mM PIPES, 25 mM HEPES, 10 mM EGTA, and 4 mM MgSO$_4$·7 H$_2$0. The cells were treated twice with 50 mM NH$_3$Cl for 15 min each for PFA-fixed cells, or 0.01 mg of NaBH$_4$ in 1× PBS for 7 min for glutaraldehyde-fixed cells to quench autofluorescence. Cells were then washed twice with PBS. Cells were permeabilized with 0.25% Triton-X 100/TBS for 10 min at RT. Cells were rinsed twice in 1× TBS for 5 min. Blocking for non-specific binding was done by incubating samples in 10% donkey serum in 0.2% Tween-20/TBS for 60 min at RT. Samples were washed twice with PBS. Primary antibodies (*Table 1*) were added to samples and incubated for 60 min at RT or 4°C overnight in dilutant made of 1% donkey serum in 0.025% Triton X-100/TBS. Samples were washed in 0.25% Triton-X 100/TBS 4–6 times, 10 min each. Secondary antibodies diluted in 1% donkey serum and 0.025% Triton X-100/TBS were incubated on samples for 60 min at RT. Samples were washed with 0.25% Triton-X 100/TBS 4–6 times 10 min, stained with DAPI (1:1000) in PBS for 5 min at RT, again washed with PBS and directly imaged or

coverslips were added on slides using ProLong Gold antifade (Thermo Fisher Scientific), according to the manufacturer's instructions. Confocal imaging was done on a Leica SP5 using the LAS-AF software, 405 nm diode, argon and 561 and 648 nm laser lines, three Photomultiplier tubes, and one HyD GaSP detector, as per the requirement of the experiment. Images were scanned using a 63× 1.4 NA oil immersion objective and later processed using ImageJ and Imaris software.

## IFT-A sequence homology search and structural modeling

The sequence match of IFT-A proteins was found by iterative rounds of homology searches via alignment for sequence proximity-based clustering as described before (*Wells et al., 2017*; *Wells and Marsh, 2019*). Further SWISS-MODEL server was used to model IFT-A complex protein structures as described on the server (*Waterhouse et al., 2018*). Briefly, a template search with BLAST and HHblits was performed against the SWISS-MODEL template library. The target sequence was searched with BLAST against the primary amino acid sequence contained in the SMTL. An initial HHblits profile, followed by one iteration of HHblits against NR20, was run and the obtained profile then searched against all profiles of the SMTL. The top hit in all of IFTA searches was 3mkqA (*Lee and Goldberg, 2010*), a coatomer β′ subunit 2.5 Å X-ray structure with 14–20% sequence identity and 25–30% sequence similarity with different IFT-A proteins. A coatomer α subunit was also found within these top matches. Models were built on the target-template alignment using ProMod3. Coordinates that are conserved between the target and the template were copied from the template to the model. Insertions and deletions were remodeled using a fragment library. Side chains were then rebuilt. Finally, the geometry of the resulting model was regularized by using a force field. In case loop modeling with ProMod3 fails, an alternative model was built with PROMOD-II. The global and per-residue model quality has been assessed using the QMEAN scoring function. The obtained model was processed later in Pymol software for structural analysis.

## Cloning, expression, and purification of the *C. reinhardtii* IFT-A trimeric (IFT139/121/43) complex from mammalian cells

The codon-optimized sequences for *C. reinhardtii* IFT-A trimeric complex (UniProt accession codes: IFT43_A8HYP5, IFT121_A8JFR3, and IFT139_A9XPA6) were assembled into a single construct for expression in mammalian cells. The *IFT43* gene was fused to TEV cleavable His-GFP-tag at the N-terminus for affinity purification and inserted into pAceCMV vector while the *IFT139* and *IFT121* subunits were untagged. The pAceCMV_His-GFP-IFT43, pIDC_IFT121, and pIDK_IFT139 were fused using an in vitro Cre recombinase (New England Biolabs) by the LoxP sites in the vectors to form the IFT-A trimer construction.

Large-scale transient expression of the IFT-A trimeric complex in mammalian HEK293S cells was carried out by transfection of the IFT-A trimer construct using PEI (40 kDa linear polyethylenimine, 1 mg/ml stock in water). Before transfection, sterile and high-quality DNA was prepared using a NucleoBond Maxiprep Kit (MACHEREY-NAGEL) with 200 ml overnight culture of DH5α cells containing the construct. HEK293S cells were cultivated 1 day before the transfection in medium (FreeStyle 293 Expression Medium, Thermo Fisher) with 1% FBS and 1% penicillin/streptomycin. Cultures were incubated in a humidified incubator with 5% $CO_2$ at 37°C with 130 rpm shaking to let the cells grow. The cells were diluted to $1.1 \times 10^6$ cells/ml before transfection with fresh and warm medium. The transfection mixtures were prepared using a sterile flow bench. For expression in 1 l of HEK293S cells, 1000 µg IFT-A trimer DNA were diluted into 25 ml medium without antibiotics or FBS. In another tube, 3000 µg PEI were diluted in 20 ml medium and added to the diluted DNA dropwise. The mixture was incubated at RT for 5 min to let the PEI-DNA transfection complex form. The mixture was added dropwise to cells and mixed gently by swirling the flask. Cells were incubated at 37°C in a $CO_2$ incubator for 48 hr. The cells were harvested by centrifugation at 800 × g for 10 min at 4°C, and the cell pellet was flash-frozen in liquid nitrogen and stored at –80°C until use.

The IFT-A trimeric complex was purified using the His-tag on the IFT43 for affinity chromatography. Briefly, a frozen pellet from 1.5 l of HEK293S cell culture was thawed on ice and resuspended in lysis buffer (50 mM HEPES pH 7.4, 250 mM NaCl, 2 mM $MgCl_2$, 10% [v/v] glycerol, and 5 mM β-mercaptoethanol) supplemented with 1 µl DNase and one cOmplete Protease Inhibitor Tablet (cOmplete-EDTA Free Protease Inhibitor Tablet, Roche Applied Science) to a final volume of 20 ml. Cells were lysed in a dounce-type tissue grinder (Wheaton) using 30 strokes. The cell lysate was

cleared by centrifugation at 48,000 × g for 45 min at 4°C. The clarified supernatant was loaded onto a 1 ml TALON column (HiTrap, Cytiva) pre-equilibrated with lysis buffer at 4°C. The bound protein was washed with 15 mM imidazole in $Q_A$ buffer (20 mM Tris–HCl pH 7.5, 10% glycerol, 50 mM NaCl, and 5 mM β-mercaptoethanol), followed by elution with 150 mM imidazole in $Q_A$ buffer. The elution containing the IFT-A proteins was loaded onto a 5 ml Q column (HiTrap Q FF, Merck-Millipore), and the bound IFT-A proteins were eluted in $Q_A$ buffer with a 50–500 mM gradient of NaCl. The elution fractions containing the IFT-A proteins were concentrated to 500 μl in a 100 kDa molecular weight cutoff concentrator (Amicon Ultracel, Merck-Millipore) for subsequent SEC on a pre-equilibrated Superose 6 Increase column (10/300 GL, Merck-Millipore) in SEC buffer (10 mM HEPES pH 7.5, 150 mM NaCl, 2 mM $MgCl_2$, 1 mM DTT, 5% glycerol). The SEC peak fractions were analyzed by SDS-PAGE and resulted in the sample used in *Figure 5B and C*.

## Cloning, expression, and purification of *C. reinhardtii* His-IFT43, His-IFT43/121, and His-IFT43/121/139 from insect cells

DNA sequences encoding for the *IFT43* with an N-terminal TEV cleavable hexa-histidine tag and untagged *IFT121* were cloned into the two expression cassettes of the pFL vector. The gene encoding for *IFT139* was cloned in another pFL vector.

The expression and purification of His-IFT43, His-IFT43/121 complex, and His-IFT43/121/139 complex was performed as previously described for the CrODA16 protein (*Taschner et al., 2017*) with the following modifications. Two recombinant baculoviruses for IFT139 and IFT43/121 were generated from separate constructs. The expression was carried out in sf21 suspension cells by co-infection with these two recombinant baculoviruses. After 3 days of incubation at 27°C, the cells were harvested by centrifugation.

The His-43, His-IFT43/121, and His-IFT43/121/139 were purified using a similar purification procedure to that for the His-GFP-tagged IFT-A trimer describe above and were purified by Ni-NTA affinity, ion-exchange, and SEC. The SEC fractions containing His-tagged IFT-A proteins were used for the lipid overlay assays in *Figure 5—figure supplement 1B–D*.

The purified His-IFT43/121 and His-IFT43/121/139 were digested overnight using TEV protease for removal of His-tag. The resulting IFT-A proteins were loaded onto SEC, and fractions containing untagged IFT-A proteins were further used for the binding assay with liposomes in *Figure 5D*.

## Lipid overlay assay

To detect the direct binding between non-core IFT-A complexes and lipids, the His-GFP-tagged IFT-A trimeric complex or His-tagged proteins purified from insect cells and Membrane Lipid Strips (Echelon Biosciences, P-6002) with 100 pmol of 15 different lipids were used following the manufacturer's protocol. The strips were blocked in 3% (w/v) BSA in TBS-T buffer (50 mM Tris [pH 7.4], 150 mM NaCl, and 0.1% [v/v] Tween 20) at 4°C overnight in dark with gentle agitation. After blocking, they were washed in TBS-T buffer three times and 5 min each, followed by incubation at RT for 1 hr with IFT-A proteins in SEC buffer supplemented with 3% (w/v) BSA. The strips were washed three times in TBS-T as before and soaked in 3% (w/v) BSA in TBS-T with primary antibody against His-tag (THE His Tag Antibody, Mouse, GenScript) at a 1:2500 dilution for 1 hr at RT. Strips were washed three times and incubated with horseradish peroxidase (HRP)-conjugated polyclonal rabbit anti-mouse immunoglobulins (1:1000 dilution, Dako) for 1 hr followed by three TBS-T washes. An ECL Prime Western Blotting reagent (Amersham) was used as the substrate for the HRP, and the binding of IFT-A proteins onto spotted lipids was recorded with the ChemiDoc imaging system (Bio-Rad).

## Negative stain EM

The POPC-liposomes and PA-containing liposomes (PE/PG/PA) were purchased from T&T Scientific Corp. The liposomes (PE/PG/PA) have a similar phospholipids composition to that of *Chlamydomonas* ciliary membrane as reported previously (*Lechtreck et al., 2013*). The percentage of PA was 11.36% while the ratios of PE and PG as the framework of liposomes were requantified to 63.18 and 25.46%, respectively.

To observe the binding between IFT-A complexes with liposomes, the liposomes (PE/PG/PA, 0.20 mM) were applied to homemade carbon grids directly or after incubation with IFT-A complexes (untagged IFT139/121/43 trimer or IFT121/43 dimer, 0.25 μM) at 25°C for 10 min. 3 μl of the sample

were applied to the plasma-cleaned grids for 30 s before it was blotted, and the sample was stained with 2% (w/v) uranyl-format staining by applying 3 μl of stain three times on the grids. The negative stain grids were imaged on an FEI Tecnai G2 Spirit TEM operated at 120 kV with a 67,000× nominal magnification corresponding to the digital pixel size of 1.59 Å/pixel. The electron micrographs were recorded on a water-cooling 4k CMOS CaMeRa (TemCam-F416). The mixture of IFT-A trimer (0.10 μM) with POPC liposomes (0.20 mM), as a negative control, was checked using negative staining EM by following the same procedure.

## Transmission electron microscopy

### TEM sample preparation

24 hr serum-starved MEFs were chemically fixed for flat embedding using the following protocol: (1) cells were grown on 60 mm dishes, and ciliogenesis was induced by serum starvation for 24 hr. (2) For prefixation under culture conditions, 25% glutaraldehyde was added to the growth medium to a final concentration of 1%, mixed gently, and incubated for a few minutes at 37°C. (3) The growth medium (containing the glutaraldehyde) was replaced with a sample buffer (0.1 M HEPES, 4 mM $CaCl_2$, pH 7.2) containing 2% glutaraldehyde and incubated 1 hr at RT (replacing the fixation buffer with a fresh one after 20 min). All prefixation solutions were prewarmed to 37°C, and all steps were done at 37°C, to preserve the cytoskeleton. (4) The fixation buffer was replaced with fresh fixation buffer and incubated for 4 hr at 4°C. (5) After that, the sample was washed once in sample buffer and 2–3 times in distilled water, each for 5–10 min, gently removing and replacing the buffer. (6) Samples were incubated in 1% $OsO_4$ (EMS) (in distilled water) for 1 hr at 4°C, (7) washed 3–4 times for 10 min each in distilled water, and (8) incubated in 1% uranyl acetate (EMS) in distilled water overnight at 4°C. (9) Then, samples were rinsed 3–4 times for 10 min each in distilled water and (10) dehydrated using a graduated series of ethanol: 30, 50, 70, 80, 90, and 96% ethanol, 5 min each step at 4°C, followed by twice rinsed in anhydrous 100% ethanol 10 min each at RT. (11) Infiltration was performed using a 1:1 mixture of LX112 (Ladd Research, USA; EMS) and ethanol 2 hr, followed by pure LX112 overnight and another 2 hr pure LX112, where all steps were performed at RT. (12) Flat embedding: for flat embedding, the caps of the BEEM embedding capsule (size, #3, EMS) were cut off and capsules filled with LX112. The capsules were inverted over a selected area of the cell monolayer in the dish, and the resin cured at 60°C oven for 48 hr. The capsule was then removed by breaking off from the dish, leaving the monolayer cells embedded in the surface of the block. (13) Sectioning and post-staining: for sectioning and post-staining, 300-nm-thick serial sections were cut by Leica Ultracut UCT (Leica Microsystems, Wetzlar, Germany) with a diamond knife and sections picked up with a Formvar (EMS) coated 1 × 2 mm slot copper grid (EMS). Sections were post-stained with 2% uranyl acetate for 10 min, then with lead citrate for 5 min. Imaging: sections were stained on the grid with fiducials (15 nm gold nanoparticles, Sigma-Aldrich). 70-nm-thick sections were cut for regular TEM imaging, and 300-nm-thick sections were prepared for tomographic acquisition.

Tilt series were acquired on a Tecnai F30 (FEI) transmission electron microscope, operated at 300 kV, and equipped with 2048 × 2048 Gatan CCD camera and FEI Titan Halo transmission electron microscope operated at 300 kV equipped with a field emission gun (FEG) and a Gatan K2 direct detector. The SerialEM software (*Mastronarde, 2005*) was used for automatic acquisition of double tilt series. Tomographic tilt series were recorded with a pixel size of 1.235 nm on Titan Halo and 1.178 nm on F30, a maximum tilt range of about 60°, and tilt steps of 1°. Tomographic reconstruction, joining of tomograms from consecutive sections, segmentation, and visualization of the tomograms, was done using the IMOD software package (*Kremer et al., 1996*). In WT cells, very few vesicles are observed around the base of the cilia that mostly have electron-dense coats with a range of intensities, including those observed fusing with the ciliary pocket or sheath. For simplicity, we have labeled these as 'coated' and colored them magenta. By comparison, the accumulation of vesicles around the base of *Wdr35* mutant cilia generally lacks prominent densities around them. For simplicity, we have labeled these as 'coatless' and colored them cyan as whether coats previously existed but disassembled or whether they failed to form in the first place is not clear from our study.

24 hr serum-starved WT, *Wdr35*[-/-], and *Dync2h1*[-/-] cells were serially sectioned parallel to the adherent surface. Two to four 300 nm parallel serial sections are required to get the whole 3D volume ultrastructural view covering full cilia and their cellular surroundings. We reconstructed 45 tomograms

to get a minimum of 3–4 whole-cell volumes for each genotype. We took micrographs of 30 WT, 20 *Wdr35^-/-^*, and 30 *Dync2h1^-/-^* cells for this study.

## Correlative light and electron microscopy

WDR35-EmGFP and ARL13B-mKate expressing *Wdr35^-/-^* MEFs were serum-starved for 24 hr, stained with Hoechst 33342 (R37605) for 10 min in culture condition, fixed with 4% PFA and 0.1% GA in 1× PHEM and imaged on Zeiss LSM 880 upright single-photon point scanning confocal system with Quasar detector (32 spectral detection channels in the GaAsP detector plus 2PMTs) and transmitted light detector, Airyscan detector for high-resolution imaging. Cells were grown on 35 mm glass-bottom dishes with grids (cat. no. P35G-1.5-1.4C-GRID) and firstly brightfield images were made with Plan-Apochromat 10×/0.45 M27 objective to save the coordinates of cells needed for the correlation with the respective TEM data. Confocal and Airyscan imaging was done using Plan-Apochromat 63×/1.4 oil DMC M27 objective, 405 nm laser diode, 458, 477, 488, 514 nm multiline integrated argon laser and 594 nm integrated HeNe laser. Z-stack was acquired sequentially to get the whole 3D volume of the cell, and the image was further deconvolved using the in-built software. After Airyscan imaging, the sample was processed for TEM as described above. 70 nm sections were made for the regions of saved coordinates from brightfield imaging, mounted on grids and imaged on FEI Morgagni TEM (100 kV) microscope.

## Immunogold labeling

*Wdr35^-/-^* MEFs expressing WDR35-EmGFP and ARL13B-mKate2 (*Table 2*) were serum-starved for 4 hr. MEF cells were grown on 6 mm sapphire disks (Wohlwend GmbH, Switzerland, 1292) and high-pressure frozen (EM ICE, Leica Microsystems). The frozen samples were processed by freeze substitution in a Leica AFS2 temperature-controlling machine (Leica Microsystems) using 0.01% uranyl acetate (Polyscience Europe GmbH, 21446) and 4% water in glass distilled acetone (EMS, E10015) as freeze substitution medium and then embedded in Lowicryl HM-20 (Polysciences, 15924-1). 70-nm-thick serial sections were sectioned on a Leica Ultracut UCT ultramicrotome (Leica Microsystems). Sections were labeled with anti GFP antibody, 1:20 (Abcam, ab6556; *Table 1*) followed by secondary goat anti-rabbit antibody coupled to 10 nm gold, 1:30 (BBI Solutions, batch 008721; Table 2). Before antibody staining, grids were incubated twice section side for 10 min each on blocking buffer PBG (0.5% BSA/0.1% fish skin gelatin in PBS). Following blocking, grids were incubated for 1 hr in primary-Ab/PBG in a wet chamber, given five 2 min washes with PBG and incubated for 1 hr in secondary-Ab/PBG. Grids were washed five times for 2 min with PBG, followed by five 2 min washes with PBS. Antibodies were subsequently fixed for 1 min 0.1% glutaraldehyde/PBS, followed by five 2 min washes with PBS and five 2 min washes with $H_2O$. After immunogold labeling, the sections were stained with 1% uranyl acetate (Polyscience Europe GmbH, 21446) in water for 8 min and 0.04% lead citrate (EMS, 17800) for 5 min. The sections were imaged using Tecnai 12 (Thermo Fisher Scientific, formerly FEI/Philips) at 100 kV with TVIPS F214 and F416 cameras (TVIPS, Gauting, Germany).

## Image analysis and measurements

All image processing were performed using Fiji (*Schindelin et al., 2012*). Macros for quantification of PCM1 (RadialIntensityFromCentrosomes.ijm) and clathrin (3DMeanIntensityfromUserDirectedPoints.ijm) can be found on GitHub (https://github.com/IGC-Advanced-Imaging-Resource/Quidwai2020_WDR35paper; *Quidwai et al., 2021*; copy archived at swh:1:rev:96b375ac31f-1451dea93943fac7f563ad348ee69). To measure PCM1 intensity radially from the centrosomes, an average intensity projection of the z-stack was obtained, and the γ-tubulin signal was segmented using RenyiEntropy threshold and the Analyze Particles tool to obtain masks of the centrosomes. The selections obtained from the masks were enlarged using the 'Make Band' function to create a band region of interest (ROI). This was done by increasing in 1 µm increments until there were five bands. The centrosome masks and the surrounding bands were measured on the PCM1 channel of the average intensity projection image. To quantify clathrin intensity around the cilia base, a point was manually selected as the center of the basal point. The user was blinded to file name and condition while quantification took place. This point was expanded 1 µm in each direction to create a shall of 2 µm diameter in x, y, and z. This shell was then measured using the 3D image suite in ImageJ (*Ollion et al., 2013*). Etomo and IMOD (*Kremer et al., 1996*) were used to reconstruct tomograms

and manually segment tomograms, respectively. These segmentations were used to create objects using the 3D Image suite in Fiji. The 3D centroids were obtained and the manually segmented ROI on the 2D slice that the 3D centroid was on was selected to move forward with. A 20 nm width band around this ROI was measured using the 'Make Band' function. The integrated density of this band ROI was quantified as an indication of how electron dense the region around the user segmented vesicle is. 3D objects were measured using the 3D Image Suite. Statistical analyses were carried out in GraphPad Prism 8.

## Acknowledgements

We thank the IGC Advanced Imaging Resource and the IGC Mass Spectrometry facility (in particular, Jimi Wills and Alexander von Kriegsheim). We thank the Electron Microscopy Facility (in particular, Tobias Fürstenhaupt, Michaela Wilsch-Bräuniger and Daniela Vorkel) and the Light Microscope Facility from the Services and Facilities of the Max Planck Institute of Molecular Cell Biology and Genetics, Dresden (in particular, Sebastian Bundschuh). We thank Rick Kahn, Toby Hurd, Ian Jackson, and Patricia Yeyati for helpful discussions and comments on the manuscript. We are grateful to Greg Pazour (UMass) and Victor Ruiz (UAM) for sharing custom antibodies. We thank Jonathan Eggen-schwiler (UGA) for Ift122::GFP and Kenji Kontani (UOT) for the ARL13B-EGFP used in this study, as well as Philip Beachy for pEGFP-mSmo (Addgene plasmid # 25395; http://n2t.net/addgene:25395; RRID:Addgene_25395).

## Additional information

### Funding

| Funder | Grant reference number | Author |
|---|---|---|
| European Molecular Biology Laboratory | Short Term Fellowship number 7961 | Tooba Quidwai |
| European Commission | H2020 Grant No. 888322 | Narcis A Petriman |
| Lister Institute of Preventive Medicine | Research Prize Fellow | Joseph A Marsh |
| Novo Nordisk | Grant No. NNF15OC0014164 | Jiaolong Wang Narcis A Petriman Esben Lorentzen |
| Carlsbergfondet | Grant No. CF19-0253 | Jiaolong Wang Narcis A Petriman Esben Lorentzen |
| European Commission | H2020 Grant No. 819826 | Weihua Leng Petra Kiesel Gaia Pigino |
| European Commission | H2020 Grant No. 866355 | Tooba Quidwai Emma A Hall Pleasantine Mill |
| Medical Research Council | Core unit funding No. MC_UU_12018/26 | Tooba Quidwai Emma A Hall Laura C Murphy Margaret A Keighren Pleasantine Mill |
| Edinburgh Super Resolution Imaging Consortium (ESRIC) | | Tooba Quidwai |

The funders had no role in study design, data collection and interpretation, or the decision to submit the work for publication.

## Author contributions
Tooba Quidwai, Formal analysis, Investigation, Methodology, Visualization, Writing – original draft, Writing – review and editing, Conceptualization, Data curation, T.Q. developed the project, performed the bulk of the experiments (including FM, TEM, tomography, and segmentation, CLEM, proteomics, homology modeling and cell culture), quantified and analysed the data, interpreted results and prepared figures; Jiaolong Wang, Investigation, J.W. expressed and purified CrIFT_A complex and performed the lipid overlay assays, as well as the liposome assays., Visualization; Emma A Hall, Formal analysis, Resources, Writing – review and editing; Narcis A Petriman, Investigation; Weihua Leng, Investigation, Methodology, W.L. prepared samples for immunoEM and EM imaging, as well as acquired and reconstructed tomographic data for the rescue experiments; Petra Kiesel, Investigation, P.K. contributed samples for immunoEM and rescue experiments; Jonathan N Wells, Investigation, Visualization; Laura C Murphy, Formal analysis, Methodology, Validation; Margaret A Keighren, M.A.K maintained mouse lines., Resources; Joseph A Marsh, Methodology, Supervision; Esben Lorentzen, E.L. contributed to data analysis and results interpretation for the in vitro studies., Methodology, Supervision, Writing – review and editing; Gaia Pigino, Funding acquisition, G.P. contributed to data analysis and results interpretation, supported and provided resources funding for the EM and CLEM part of the project., Methodology, Resources, Supervision, Writing – review and editing; Pleasantine Mill, Conceptualization, Data curation, Formal analysis, Funding acquisition, Investigation, P.M. conceived and supervised the project, contributed to data, its analysis and results interpretation, as well as provided resources and funding., Resources, Supervision, Writing – original draft, Writing – review and editing

## Author ORCIDs
Tooba Quidwai ⓘ http://orcid.org/0000-0001-5248-9010
Narcis A Petriman ⓘ http://orcid.org/0000-0002-3189-7530
Jonathan N Wells ⓘ http://orcid.org/0000-0003-3700-020X
Esben Lorentzen ⓘ http://orcid.org/0000-0001-6493-7220
Pleasantine Mill ⓘ http://orcid.org/0000-0001-5218-134X

## Ethics
We followed international, national and institutional guidelines for the care and use of animals. Animal experiments were carried out under UK Home Office Project Licenses PPL 60/4424, PB0DC8431 and P18921CDE in facilities at the University of Edinburgh (PEL 60/2605) and were approved by the University of Edinburgh animal welfare and ethical review body.

## Decision letter and Author response
Decision letter https://doi.org/10.7554/eLife.69786.sa1
Author response https://doi.org/10.7554/eLife.69786.sa2

# Additional files

## Supplementary files
• Transparent reporting form

## Data availability
Source Data (Figures 1B,C; 2B; 3B; as well as Figure 3-Figure Supplement 1C, Figure 7-Figure Supplement 1C) have been uploaded with the submission containing numerical data of all graphs shown in the figures and figure supplements. We have also uploaded the Excel or/and Prism files as source data in addition to the data points which have been referenced, as appropriate in the Figure legends. Source data of raw and full uncropped blots for Figures 3B, C, E and Figure 3-Figure Supplement 1A, as well as Figure 5B, C and Figure 5-Figure Supplement 1 B-D are uploaded as zipped files per figure. For Figure 7D, we have included the ROIs used for calculations uploaded in a single folder including all numerical data of Figure 7 (7B-D) graphs at Dryad https://doi.org/10.5061/dryad.m37pvmd33. All analysis tools have been made available on GitHub (https://github.com/IGC-Advanced-Imaging-Resource/Quidwai2020_WDR35paper, copy archived at https://archive.softwareheritage.org/swh:1:rev:96b375ac31f1451dea93943fac7f563ad348ee69), as described in Materials and Methods. Proteomics data files are uploaded to ProteomeXchange (Identifier: PXD022652). Project Name: A

WDR35-dependent coatomer transports ciliary membrane proteins from the Golgi to the cilia Project accession: PXD022652.

The following dataset was generated:

| Author(s) | Year | Dataset title | Dataset URL | Database and Identifier |
|---|---|---|---|---|
| Quidwai T, Hall EA, Mill P | 2021 | A WDR35-dependent coatomer transports ciliary membrane proteins from the Golgi to the cilia | https://www.ebi.ac.uk/pride/archive/projects/PXD022652 | PRIDE, PXD022652 |
| Mill P, Quidwai T, Murphy L, Pigino G | 2021 | Figure 7- source data 1 | https://doi.org/10.5061/dryad.m37pvmd33 | Dryad Digital Repository, 10.5061/dryad.m37pvmd33 |

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
