## [Editor Report]

This work supports a model in which the intraflagellar transport complex, IFT-A, acts as a coat to support Golgi to cilia traffic of ciliary cargos. Data also show that proteins in the IFT-A complex are co-dependent and bind to specific lipids, consistent with such a coat function. Consistent with this model, in the absence of one core component of IFT-A, IFT122/WDR35, uncoated vesicles accumulate at the base of cilia, suggesting a requirement for the coat in fusion at cilia.

---

## [Decision Letter]

Thank you for submitting your article "A WDR35-dependent coat protein complex transports ciliary membrane cargo vesicles to cilia" for consideration by *eLife*. Your article has been reviewed by 3 peer reviewers, including Richard A Kahn as the Reviewing Editor and Reviewer #1, and the evaluation has been overseen by Vivek Malhotra as the Senior Editor.

All three reviewers found sufficient merit and impact from the work described to warrant publication. A number of strengths were identified by each of the reviewers, that include the quality of much of the data, the use of some powerful techniques to address a challenging problem, furthering the homology model of IFT and COP complexes, the dependence of IFT-A complexes (but not IFT-B) on WDR-35/IFT121, and identification of at least two morphologically distinct types of vesicles linked to ciliary traffic or defects. This manuscript will clearly add to our nascent understanding of IFT-A actions in cells and at cilia, while also opening up new lines of investigation that will further add to our understanding of them.

There was also consensus that conclusions as to the origins of the vesicles identified were over-stated or at least less clear than suggested in the text. Inclusion of a control in the tomographic study of WDR35 -/- MEFs appears necessary. Similarly, inclusion of a rescue experiment of these cells would further strengthen the data and conclusions. And the singular identification of PA as the lipid to which IFT-A complexes bind also would be strengthened by additional data, or at least qualifiers in the text regarding the limits of the sole assay employed and interpretations. A number of suggestions to strengthen conclusions and extend the data and new conclusions are offered. At a minimum the authors are encouraged to edit the manuscript with these reviews in mind so as to not over-interpret the current data, specifically including adding qualifying statements or limitations of the data presented, and consider performing some of the proposed experiments, as well as edits identified for accuracy or enhanced presentation. A number of specific edits to figures and legends should also be made, including possible replacing some of the cell imaging data.

Essential revisions:

1. The origins of the vesicles identified were over-stated or at least less clear than suggested in the text. Inclusion of a control in the tomographic study of WDR35 -/- MEFs appears necessary. inclusion of a rescue experiment of these cells would further strengthen the data and conclusions

2. The singular identification of PA as the lipid to which IFT-A complexes bind also would be strengthened by additional data, or at least qualifiers in the text regarding the limits of the sole assay employed and interpretations.

3. A number of suggestions to strengthen conclusions and extend the data and new conclusions are offered. At a minimum the authors are encouraged to edit the manuscript with these reviews in mind so as to not over-interpret the current data.

4.A number of specific edits to figures and legends should also be made, including possibly replacing some of the cell imaging data. See the reviewers comments on specific figures.

*Reviewer #1:*

The manuscript from Quidwai, et al., "A WDR-35 dependent coat protein…" uses a number of approaches in efforts to define novel roles for IFT-A subunits, most notably WDR-35/IFT-121, in vesicle traffic to growing, primary cilia. The manuscript contains a bit of a technical tour de force, employing knockout (WDR-35 vs the retrograde motor dynein heavy chain, Dync2h1) cell lines (MEFs), fixed immunofluorescence and transmission EM imaging, Western blotting and mass spectrometry to characterize abundance of protein subunits in complexes and in entire cells, iterative homology searching of protein alignments in silico, and lipid binding assays with purified recombinant proteins. Together they are used to build the case that WDR-35/IFT121, an IFT-A accessory (as in non-IFT-A core subunit) and by extension IFT-A forms a lipid binding coat complex that is needed to transport ciliary proteins into cilia. And in the absence of WDR-35, instead the cilia are formed by stunted in growth and instead accumulate uncoated vesicles, defective in fusion at the ciliary pocket. The bulk of the data go a long way towards convincing this reader of the bulk of the conclusions, though with some reservations. Among the most interesting observations in the paper are the consequences of WDR35/IFT121 deletion on ciliary length and stability (or lack thereof) of other IFT-A, but not IFT-B subunits, that there exists some structural and likely functional homologies between IFT and COP complexes, and that cells lacking WDR-35/IFT121 accumulate uncoated vesicles, consistent with IFT-A complexes forming that missing coat. In addition, there is extensive citation of the literature from studies across multiple species that overall does a very good job of putting the work into perspective. I believe that this work contributes to the field resulting in a new and better, though still quite incomplete, model of the roles of IFT-A in cilia.

I have a number of very minor concerns that can be readily addressed but one larger one. Specifically, the authors are clearly concluding that the source of the uncoated vesicles seen in TEM imaging in WDR-35/IFT121 KO cells is Golgi. I see no strong evidence to support this and suspect it is based upon the homologies to COPI and its actions there (?). While the title carefully avoids mention of the source of such vesicles, I think the manuscript would be stronger with a clearer statement as to evidence supporting the origins and perhaps the level of uncertainty on this issue. The paragraph beginning on line 344 appears an effort to make this case but could be stronger.

I have a number of very minor concerns that can be readily addressed but one larger one. Specifically, the authors are clearly concluding that the source of the uncoated vesicles seen in TEM imaging in WDR-35/IFT121 KO cells is Golgi. I see no strong evidence to support this and suspect it is based upon the homologies to COPI and its actions there (?). While the title carefully avoids mention of the source of such vesicles, I think the manuscript would be stronger with a clearer statement as to evidence supporting the origins and perhaps the level of uncertainty on this issue. The paragraph beginning on line 344 appears an effort to make this case but in my view it fails.

Also, the lipid binding assay is one result, showing binding specificity of purified recombinant proteins to phosphatidic acid. While intriguing/interest and apparently quite specific some question the accuracy of this assay when used as the sole evidence for such lipid specificity. And perhaps more discussion of sources of PA and why it might make sense would help the reader. Finally, I end with some experiments that one might propose to further strengthen this manuscript. While I don't support withholding publication until done, they might be worth considering now or later. These might include following up the lipid strip binding assays with assays employing lipid vesicles of distinct composition. Similarly, the TEM could be furthered with immunogold labeling to nail down whether the coats seen in WT vesicles are studded with IFT-A proteins and absent in the KO.

*Reviewer #2:*

Wdr35 has been previously shown to be trafficking diverse set of ciliary membrane proteins including smoothened, GPCRs and Arl13b (Fu et al., 2016). The authors repeat and extend some of these observations and further show that IFT-A subunits have close sequence and structural similarity to COPI subunits. The authors show purification of IFT-A peripheral subunits that together bind to phosphatidic acid in lipid-strip assays. Using TEM techniques they observe electron dense vesicles tracking between Golgi and ciliary base, whereas lack of Wdr35 shows abundant vesicles with less density. Lack of Dynein 2, a retrograde ciliary motor, shows paucity of vesicles around ciliary base. Two WDR35-positive vesicles are shown to be electron dense in Wdr35 null cells by correlative light and electron microscopy. The authors suggest that Wdr35 and likely other IFT-A subunits assist in cargo transport from Golgi into cilia at the stage of cilia elongation.

The authors use state-of-the-art cell biology, biochemical purification of IFT-A peripheral complex and TEM/correlative light and electron microscopy techniques to address cargo targeting to cilia.

The authors use electron density of vesicles to suggest for presence or lack of a coat in vesicles destined for cilia around ciliary base. The proposed model will be further solidified by quantifying the WDR35-GFP rescue experiments and showing quantification of coated vs noncoated vesicles in their TEM and CLEM assays.

Lack of vesicles in wild type vs Dynein 2 knockout cells suggest that at least some of the vesicles around ciliary base can result from post ciliary trafficking out of cilia. The abundant "preciliary" vesicles rather than being cargo carrying and destined for cilia might also include recycling, post ciliary vesicles or vesicles without any relevant cargoes.

It might also be relevant to include other interpretations for the current data of vesicle accumulation seen in Wdr35 knockout in EM. The apparent increase in all kinds of vesicles at ciliary base in the Wdr35 knockout cells could be an indirect consequence of Wdr35's role in processes independent of generating vesicular coat. For e.g., Wdr35 could promote delivery of selective cargoes to cilia through the F-actin rich ciliary base (Fu et al., 2016).

The authors use electron density on vesicles for suggesting vesicular "coat" in the current paper that is reduced in Wdr35 knockout cells, but it is presently unclear on what constitutes this coat other than Wdr35.

The authors need to solidify the data with WDR35-GFP rescues, by including data from GFP lacking vesicles (internal control) or at least showing total distribution of number, density, size for vesicles as shown for knockout MEFs.

The authors show nicely the quantification of all vesicles in figure 7B-C, but the distribution in coated vs noncoated vesicles should be shown as well.

Any cargo that can be immunolocalized to these vesicles might be a way to address concerns regarding specificity of coat containing cilia destined cargoes. Alternatively, the authors might use co-localization between WDR35-GFP and prospective cargoes (e. g. Arl13b-mKATE-2) in their CLEM methods.

Considering that the coat might result from Wdr35 associated proteins, it might be relevant to phenocopy Wdr35 knockout with respect to cells lacking other subunits of IFT-A complex to solidify the coat hypothesis. Lack of another peripheral subunit Ift139 does not cause lack of Smoothened or Arl13b in cilia (Hirano et al., 2016). Thus, it might be relevant to test or speculate on some of these other constituents.

1. Figure 3. Data in A of reduction in some core subunits in knockout cells is not supported by western blots in C, D. Increase in Ift122 is mentioned in text for the knockout but is shown to be not significant in quantification in (D).

2. Figure 5C. Please note that phosphatidic acid is an extremely transient lipid and is also present in other organelles such as ER. The lipid-strip assay interactions could be confirmed by liposome assays. Other core subunit complexes could be tested in the assay.

3. Figure 6. Please mention how the differentiation between ciliary sheath from pocket was done.

4. Figure 7-Suppl 1. In Wdr35 knockout MEFs, I see an abundance of all types of vesicles (color code not clear). These other types of vesicles including coated ones should be quantified.

5. Figure 7F. Integrated density nomenclature is confusing (vesicles have low density in wild type). The data shown are inclusive of all types of vesicles (coated, non-coated) in this quantification and with a wide disparity between total numbers (31 vs 508) between genotypes.

6. Please mention methods of how WDR35-GFP was used to rescue expression in knockout MEFs.

*Reviewer #3:*

The authors set out to understand why WDR35, an IFT-A component, affects ciliary length. They demonstrated that WDR35 affects non-core IFT-A degradation and ciliary localization, important for ciliary length control. Using structural predictive analysis WDR35 and IFT-A components were found to have homology to the COPI coatomer proteins. Using fixed and live cells imaging the authors demonstrated that WDR35 affects membrane cargo transport to the cilium. The authors impressively used quantitative EM tomograph studies to demonstrate that vesicles at the cilia base were uncoated and appeared to fail to fuse to ciliary membranes in WDR35-/- cells. However, whether there is a direct connection between WDR35 function as a coat protein in vesicle trafficking remains to be seen. As the authors indicated, there could be other explanations for this observation. Overall, this work has a high degree of significance to the poorly understood mechanisms of ciliary vesicle trafficking important in ciliogenesis and ciliary signaling. It is the first study to provide evidence that an IFT-A/WDR35 coatomer could function in delivering ciliary membrane cargo needed for cilia elongation.

Points to help strengthen the conclusions of this manuscript:

1. Inclusion of controls (non-ciliating) for WDR35-/- would help establish if the uncoated vesicles near the base of the cilia observed by electron microscopy are in fact associated with ciliogenesis. It is interesting that Arl13b displays strong PM/cytoplasmic staining in the absence of WDR35-/- (Video 2). Why do you think this is and could this suggest a more global effect on membrane trafficking in the cell which might explain the presence of more uncoated vesicles near the basal body?

2. The CLEM-based evidence that WDR35 localizes to vesicles is also not overly convincing for vesicular association of WDR35. Since TEM sections were used as opposed to more 3D tomography it would be expected to be challenging to ensure the vesicles identified correspond to the fluorescence signal. It is also not clear how many CLEM data sets the author performed which shows clear localization to coated vesicles. Live cell imaging and/or colocalization with other vesicles trafficking markers could provide supporting evidence that WDR35 is associated with vesicles.

3. I have some concerns about conclusions made based on ciliary localization and trafficking into the cilium in WDR35-/- cilia, in particular the fact that the Actub marker used for the cilia does not always stand out from other cytoplasmic microtubules. While I appreciate that >80% cells should have a short cilia in WDR35-/- it is not always clear where the cilia is in fluorescence imaging studies. There is considerable variability of Actub specific staining in cells. Issues with this and other figures are as follows. In Figure 4C for all 3 images representing WDR35 -/- conditions it is just not possible to identify precisely where the cilia is located, although it appears there is no accumulation of the lipid marker in any of the structures shown. There is a similar issue with Figure 3 Supplement 1 C in determining the length of the cilia being shown given the high non-ciliary signal observed for Actub. Figure 1D the authors indicate that the WDR35-/- cells are not polyglutamylated. It would be more convincing if another cilia marker was included since there is no quantification of the data provided or a centrosome marker was used. The conclusion reached about IFT-A components (Figure 3F) in WDR35-/- being in the transition zone seems hard to reach without the costaining with TZ markers or use of super resolution microscopy. In the case of Palmy-Myr-GFP localization for control and WDR35-/- cells, the later looks more like Golgi while the wt seems more vesicular. Thus, the authors conclusions about accumulation at the basal body of WDR35-/- is not well supported by the images shown in Figure 4C in my opinion. Overall, I find the fluorescence images could be better presented. In most cases too much space is taken up to show a field of view containing non-ciliary regions of the cell, while the centriole/cilia region are barely enlarged in cropped images. As quantification of some ciliary markers (ie TZ) was not done it might be better to show enlarged cropped areas of multiple cilia for representation.

4. Given the number of full cell/cilia tomographs I wonder if the assertion that only 1% of cilia have fused with the plasma membrane is correct. In Video 4 it appears as though the tip of the cilium is possibly outside the cell. The percentage of outside cilia could be better evaluated by In and out assays (Kukic et al., 2016).

[Editors' note: further revisions were suggested prior to acceptance, as described below.]

Thank you for resubmitting your work entitled "A WDR35-dependent coat protein complex transports ciliary membrane cargo vesicles to cilia" for further consideration by *eLife*. Your revised article has been evaluated by Vivek Malhotra (Senior Editor) and a Reviewing Editor.

The manuscript has been improved but there are some remaining issues that need to be addressed, as outlined below:

We believe that the requested changes may be addressed through re-wording and edits to text improve the clarity and avoid over- or mis-interpretation and that additional data are not required, though of course may further support statements made, particularly as described by reviewer #2.

*Reviewer #2:*

The authors have responded to my comments, particularly with respect to IFT peripheral subunits coating liposomes with specific lipid compositions. I do appreciate the novelty and the technical difficulties in carrying out TEM and CLEM experiments in the Wdr35 knockout cells and rescues, respectively. The additional Dynein 2 knockout TEM data is especially informative in suggesting that the accumulation of periciliary vesicles in Wdr35 knockout cells is unrelated to retrograde transport in cilia. However, I still have a few comments regarding these experiments and the interpretations intended for the general reader.

First, the authors have mentioned in the rebuttal that the color coding for vesicles in Figure 7A is by genotype. However, the color code keys suggest otherwise. It is confusing as all vesicles in Wdr35 knockout are not "uncoated". The authors have clarified in the rebuttal that they would be uncomfortable showing the distribution of the different vesicle types, rather they quantify electron density around periciliary vesicles to show significant differences between genotypes. Thus, the terms "coated vesicles" and "coatless vesicles" in Figure 6, 7A and 9 are misleading and should be changed.

Second, I might be wrong, but I don't see how the authors show by "inclusion of a wider field" in the tomographic studies of Wdr35 MEFs that the "vesicle accumulation is very regionalized over the 'thwarted' cilium". In the mentioned movies (Video 6, 7), I only see the predominant region around mother and daughter centrioles. To me, it is difficult to ascertain vesicles throughout an individual cell or in adjacent cells.

Third, quantification of periciliary vesicles, even from the limited number of cells observed from the single CLEM dataset as shown in Video 9, would really solidify the main message of the paper regarding accumulation of vesicles in the Wdr35 knockout.

Fourth, given the limited data shown for CLEM for rescued cells in Figure 8A, the vesicles pointed as #1 and #2 in the CLEM data do not convincingly represent, at least to me, the vesicles pointed to in the adjacent EMs. A Video for the CLEM cell shown in Figure 8B can be included for clarity.

Fifth, for the "electron dense vesicles" #1 and #2 shown in Figure 8A-B and the three "coated" vesicles shown to be "fusing" (apparently from another rescued cell) in Figure 8C, how are the authors sure that they are not seeing clathrin-coated vesicles to be undergoing endocytosis? Clathrin coated vesicles endocytosing are also seen in Wdr35 knockout cells in Video 7. Are similar "fusions" seen in the wild type cells for the "coated" vesicles?

Finally, the new immunogold experiments with Wdr35-GFP are not very informative and need additional controls. As the authors mentioned in the text, Wolfrum et al. (2010) showed multiple IFT-A and IFT-B proteins to localize to vesicles adjacent to the photoreceptor connecting cilia. Other IFT proteins will probably show similar pattern in the context of primary cilia.

*Reviewer #3:*

The authors have added immunoEM studies to address my concerns about vesicle localization of WDR35 which is a commendable addition to this study. One suggestion is to remove the pink dots obscuring the vesicles in the EM image or use another annotation to show what structures the gold particles are associated with vesicles.

For the revised work it still remains undetermined if WDR35 requirements in vesicle trafficking are only associated with cilia. While I appreciate this is not the focus of this study, but I still feel this analysis could demonstrate more specifically whether the defect observed in vesicle accumulation is associated with cilia or not. But I appreciate the issues that the authors may have with getting additional EM, and the problems with having cells that were not ciliated. Lastly, I appreciate the addition of the rescue studies which support a role for WDR35 in regulating vesicle traffic near the cilia.

---

## [Author Response]

Essential revisions:1. The origins of the vesicles identified were over-stated or at least less clear than suggested in the text. Inclusion of a control in the tomographic study of WDR35 -/- MEFs appears necessary. inclusion of a rescue experiment of these cells would further strengthen the data and conclusions

The reviewers raise the concern that the origin, be it Golgi or other donor membrane, of the accumulated vesicles and coated vesicles requires further investigation to nail down. These are extensive studies beyond the scope of this manuscript but will be pursued in the future. In the interim, we have toned down the text accordingly throughout to reflect this uncertainty. See lines 406, 442 and 519 of the marked revision.docx, for example.

As a control for mutant cells under the same ciliation conditions as our wild type and *Dync2h1* controls, we have included an increased field of view of the tomogram (new Video 7) of the original segmented tomogram of the periciliary vesicles which fail to fuse with the mutant ciliary pocket and ciliary sheath (Video 6). This captures not only a larger view of the original ciliating cell, but also the edges of two adjacent mutant cells grown under the same conditions. It is very apparent that the vesicle traffic defect is spatially restricted to the ‘thwarted’ mutant cilium, whilst the periphery of mutant cells is not affected (see new Figure 6—figure supplement 3). This supports that the defects observed in *Wdr35* mutant cells are not a global membrane traffic defect, but rather a focal disruption of targeted vesicles to developing the cilium. A statement has been added (lines 486-489) to make this clearer to readers and we thank reviewers for this helpful suggestion.

We have extended our analysis of the WDR35::EmEGFP rescue of the vesicle traffic defects of *Wdr35* mutants with tomograms of new cells showing reappearance of electron-dense coated vesicles and reversal of the vesicle pile-up observed in mutants (new Video 9). Accordingly, we even capture an electron-dense vesicle fusing at the ciliary base of rescued cell (new Figure 8C, Figure 8—figure supplement 1). Given limitations of time, lab moves and man power, we were unfortunately unable to segment these tomograms but they are very clear as to the reversal of mutant phenotypes. Segmentation is done manually and is very onerous. We hope the reviewers are satisfied with this expanded qualitative analysis, instead of a quantitative one.

2. The singular identification of PA as the lipid to which IFT-A complexes bind also would be strengthened by additional data, or at least qualifiers in the text regarding the limits of the sole assay employed and interpretations.

We agree completely with the reviewers. We are excited to include specific composition liposome binding assays in our revised work. Here we show that purified IFT-A non-core trimer binding specifically to liposomes composed of PE/PG/PA but not to POPC controls (see new Figure 5D). From this liposome assay, we speculate that IFT139 binding to the IFT121/43 dimer increases the binding affinity to lipids. Indeed, on its own, the IFT-A dimer signal is below the threshold of detection in the protein-lipid overlay assays but observed to weakly associate to PA-containing liposomes. Unfortunately, our attempts to better visualize structure through cryoEM of the purified IFT-A trimer with lipids have stumbled as the trimer disassembled on the cryo-EM grids during girds vitrification as the dramatically increased water-air interface, while the negative staining EM gave too low resolution images to enable us getting a higher resolution binding model. We continue to work to troubleshoot this by preventing the sample from disassembly using crosslinking and cryo-EM grids with supported film.

This is in addition to the lipid strip overlays, showing specific interactions to PA, as well as a lesser extent PS (see Figure 5—figure supplement 1D). These are all negatively charged (anionic) phosphoglycerates and importantly the trimer shows no binding to neutral or inositol-based lipids as had been reported for the IFT-A adaptor TULP3 (Mukhopadhyay et al., 2010). We have expanded our discussion of the biological importance of PA and possible implications supporting this being a novel and distinct mechanism for IFT-A membrane association.

3. A number of suggestions to strengthen conclusions and extend the data and new conclusions are offered. At a minimum the authors are encouraged to edit the manuscript with these reviews in mind so as to not over-interpret the current data.

We are very grateful to our reviewers for their very insightful comments and constructive suggestions on how to improve how we share the data, better emphasize our findings and strengthen our arguments. We have taken most of them onboard where possible.

4.A number of specific edits to figures and legends should also be made, including possibly replacing some of the cell imaging data. See the reviewers comments on specific figures.

We thank the reviewers for these suggestions. We have made edits to figures and legends, where possible to improve legibility and accessibility across platforms. See Figure 2A, Figure 3C, Figure 3—figure supplement 1A, Figure 5A. Unfortunately for the quantification of cilia length in Figures 2A and 3F, as highlighted by reviewer 3, we are limited in mouse cells to availability of robust antibodies in a third species for co-localization studies (i.e. here we use AcTub (mouse) and IFT antibodies (rabbit)). We have added additional transition zone images as requested by reviewer 3.

To supplement our CLEM data, we have included immunoEM data showing gold-conjugated anti-GFP shows sparse but specific labelling in close proximity at the ciliary base and along the ciliary sheath at putative vesicles in *Wdr35::EmEGFP* rescued *Wdr35* mutant MEFs as well as along the axoneme (see new Figure 8—figure supplement 2). We continue to work to improve our strategies and protocols to better visualize sparse staining such as IFT-As whilst retaining both membrane and microtubule ultrastructures in mammalian cells.

Reviewer #2:[…] The authors need to solidify the data with WDR35-GFP rescues, by including data from GFP lacking vesicles (internal control) or at least showing total distribution of number, density, size for vesicles as shown for knockout MEFs.

While we agree with the reviewer it would be ideal to have quantitative data on the rescue of vesicle number, density and size, this requires manual segmentation of the additional tomograms which we have generated for this dataset. It is extremely labour intensive and unfortunately, we do not currently have the hands or time for this work due to lab moves and contracts. In the revised version, we have included extensive additional high quality qualitative data we hope will satisfy the reviewer including tomograms of additional rescued cells showing reversal of vesicle accumulation phenotypes, reappearance of coated vesicles including ones that fuse in addition to restoration of ciliogenesis (see new Figure 8C, Figure 8—figure supplement 1 and Video 9).

The authors show nicely the quantification of all vesicles in figure 7B-C, but the distribution in coated vs noncoated vesicles should be shown as well.

We gently remind the reviewer that whilst beautiful quality, our datasets are from chemically fixed in-situ electron microscopy and this type of analysis is normally done exclusively on cryoEM samples. The reviewer is correct there is a mix of densities on vesicles in controls which is why we set-out ambitiously to non-biasedly call densities on all segmented vesicles in our chemically fixed samples, as shown in Figure 7F. This panel captures imperfectly the distribution of densities on all vesicles between genotypes but we felt the best way to present the data. This coated versus noncoated breakdown analysis of vesicles would be best suited to future cryoEM studies.

Any cargo that can be immunolocalized to these vesicles might be a way to address concerns regarding specificity of coat containing cilia destined cargoes. Alternatively, the authors might use co-localization between WDR35-GFP and prospective cargoes (e. g. Arl13b-mKATE-2) in their CLEM methods.

While non-ideal in resolution, there is some overlap with ARL13B and WDR35::EmGFP signal in our CLEM data in Figure 8A. We agree completely with the reviewer there is room for improvement and are actively continuing to look for ciliary cargos co-localizing with these coated vesicles. The technical challenges remain that: (a) these vesicles are relatively small and (b) labelling density of many cargos or putative coats is sparse compared to the cilia proper. The latter causes issues as the super-resolution necessary to capture these structures with modalities like STED bleach much faster than the more densely labelled cilia compartments.

Considering that the coat might result from Wdr35 associated proteins, it might be relevant to phenocopy Wdr35 knockout with respect to cells lacking other subunits of IFT-A complex to solidify the coat hypothesis. Lack of another peripheral subunit Ift139 does not cause lack of Smoothened or Arl13b in cilia (Hirano et al., 2016). Thus, it might be relevant to test or speculate on some of these other constituents.

Again, we absolutely agree with this savvy reviewer. Future studies will look to expand our analysis to other IFT-A subunit mutants to further solidify our coat hypothesis, particularly core as well as additional non-core mutants. In vertebrates, IFT139 is enigmatic as it has had a duplication, such that *Thm1* and *Thm2* may have distinct and overlapping functions to explain some of these differences reported in mammalian cells (Wang et al. 2020 *FASEB J*). In contrast, in *Chlamydomonas*, *ift139/fla17* temperature sensitive mutants show decreased flagellar levels of bacterial ion channel rhodopsins at permissive temperatures and loss of flagella at non-permissive temperatures where the rhodopsins mislocalize around the basal bodies (Awasthi et al. 2016 *Sci Reports;* Iomini et al. 2009 *Genetics*). There is lots of scope to explore some of these exciting questions in future studies.

1. Figure 3. Data in A of reduction in some core subunits in knockout cells is not supported by western blots in C, D. Increase in Ift122 is mentioned in text for the knockout but is shown to be not significant in quantification in (D).

We clarify that the IFT140 IP data in Figure 3A shows that less core IFT-A is pulled down in mutants as quantified by mass spectrometry, as well as no detectable non-core components in *Wdr35* mutants. Representative immunoblots of these IPs are shown in Figure 3B, showing similar trends that significant defects exist in assembly of IFT-A complexes without WDR35. The total protein immunoblots (Figure 3C) and quantification (Figure 3D) represent the total levels of IFT-A components showing that total levels of individual core components are unchanged however their assembly into core IFT-A complexes is compromised without WDR35. In contrast, the non-core components are unstable without WDR35, independent of a complex. The increased levels of IFT122 were observed specifically in the MEF samples as shown in Figure 3 Figure Supplement 1A, through its 4 biological replicates however not in embryo extracts shown in the main figures. We apologize for the missed figure reference in the text now rectified.

2. Figure 5C. Please note that phosphatidic acid is an extremely transient lipid and is also present in other organelles such as ER. The lipid-strip assay interactions could be confirmed by liposome assays. Other core subunit complexes could be tested in the assay.

As stated above, we agree completely with the reviewer. We are excited to include specific composition liposome binding assays in our revised work. Here we show that purified IFT-A non-core trimer binding specifically to liposomes composed of PE/PG/PA but not to POPC controls (see new Figure 5D). From this liposome assay, we speculate that IFT139 binding to the IFT121/43 dimer increases the binding affinity to lipids. Indeed, on its own, the IFT-A dimer signal is below the threshold of detection in the protein-lipid overlay assays but observed to weakly associate to PAcontaining liposomes. Unfortunately, our attempts to better visualize structure through negative EM or cryoEM of the purified IFT-A trimer with lipids have stumbled as the trimer disassembled on the cryo-EM grids during girds vitrification as the dramatically increased water-air interface, while the negative staining EM are with too low resolution images to give us to a higher resolution binding model. We continue to work to troubleshoot this. The Lorentzen group also continues to valiantly try to isolate the entire IFT-A complex and despite their extensive expertise in IFT structures are struggling to solubilize the 6 component complex, which is why we have focused on the soluble noncore trimer for this current study.

3. Figure 6. Please mention how the differentiation between ciliary sheath from pocket was done.

For this manuscript, the differentiation was solely based on the curvature of the ciliary pocket. We continue to try to optimize markers which can be used in mouse cells to better molecularly define these membrane compartments and vesicles.

4. Figure 7-Suppl 1. In Wdr35 knockout MEFs, I see an abundance of all types of vesicles (color code not clear). These other types of vesicles including coated ones should be quantified.5. Figure 7F. Integrated density nomenclature is confusing (vesicles have low density in wild type). The data shown are inclusive of all types of vesicles (coated, non-coated) in this quantification and with a wide disparity between total numbers (31 vs 508) between genotypes.

The segmentation was done manually focusing on vesicles close to the cilium and basal bodies only and the reviewer is correct that the color coding was by genotype as opposed to electron-density, and is thus subjective. To overcome bias, we undertook an automated postacquisition analysis pipeline where segmented vesicles on the raw EM images to quantify integrated density in 20 nm ring around every segmented object in 3D. This was imperfect as objects were manually drawn and sometimes crossed the membrane of the vesicles and in cases of the mutants where there were very dense vesicles that 20 nm band would overlap with other vesicle membranes. However, it was automatic and the best we could do with our chemically fixed datasets.

The discrepancy in numbers reflects how few vesicles are observed in control cell tomograms as compared to similar numbers of mutant cells. Integrated density where the EM images have a black value of 0 and white value of 255, which is the inverse of what we would expect for a fluorescent signal but we felt best to leave as is.

6. Please mention methods of how WDR35-GFP was used to rescue expression in knockout MEFs.

Please see the Material and Methods section of ‘Electroporation of MEFs’ from line 812 with details of the plasmids used in Table 3 for details on how rescue experiments were performed.

Reviewer #3:Points to help strengthen the conclusions of this manuscript:1. Inclusion of controls (non-ciliating) for WDR35-/- would help establish if the uncoated vesicles near the base of the cilia observed by electron microscopy are in fact associated with ciliogenesis. It is interesting that Arl13b displays strong PM/cytoplasmic staining in the absence of WDR35-/- (Video 2). Why do you think this is and could this suggest a more global effect on membrane trafficking in the cell which might explain the presence of more uncoated vesicles near the basal body?

The question of non-ciliating *Wdr35* controls would have been another huge experiment with additional optimization steps as it appears all of the cells we captured were undergoing ciliogenesis. Importantly, they were also matched with *Dync2h1* and wild type cells grown under the same conditions, which we would also have had to repeat to capture an earlier or different time point/condition. We believe the reviewer is asking about whether the defects in vesicle traffic are due to failure in ciliogenesis or changes in more global membrane traffic. An alternate control to address this in mutant cells under the same ciliation conditions as our wild type and *Dync2h1* controls is an increased field of view (new Video 7) of the original segmented tomogram of the periciliary vesicles which fail to fuse with the mutant ciliary pocket and ciliary sheath (Video 6). This captures not only a larger view of the original ciliating cell, but also the edges of two adjacent mutant cells grown under the same conditions. It is very apparent that the vesicle traffic defect is spatially restricted to the ‘thwarted’ mutant cilium, whilst the periphery of mutant cells is not affected (see new Figure 6 Figure Supplement 3). This supports that the defects observed in *Wdr35* mutant cells are not a global membrane traffic defect, but rather a focal disruption of targeted vesicles to developing the cilium. A statement has been added (lines 486-489 of marked revision.docx) to make this clearer to readers and we thank reviewers for this helpful suggestion. The plasma membrane accumulation of overexpressed ARL13B likely reflects the high level expression without compartment to traffic it to. In wild type cells, when ARL13B is over expressed we can also occasionally detect low level plasma membrane expression (whilst the bulk is in the cilium) suggesting it is a default when normal ciliary traffic mechanisms are overwhelmed or non-functional.

2. The CLEM-based evidence that WDR35 localizes to vesicles is also not overly convincing for vesicular association of WDR35. Since TEM sections were used as opposed to more 3D tomography it would be expected to be challenging to ensure the vesicles identified correspond to the fluorescence signal. It is also not clear how many CLEM data sets the author performed which shows clear localization to coated vesicles. Live cell imaging and/or colocalization with other vesicles trafficking markers could provide supporting evidence that WDR35 is associated with vesicles.

There is a single CLEM dataset used in the paper using the pseudo-super resolution 3D AiryScan on cells grown on grids which were then used for serial TEM sections through the entire cell. These experiments were carried out by the first author on a short-term visit to Dresden MPICBG where the TEM was done. The Mill lab continues to try to optimize markers of different membrane and vesicular compartments with WDR35 for co-localization studies, as well as endogenous tagging strategies for CLEM, as we agree this is important for future work. In the interim, to supplement our CLEM data, we have included immunoEM data showing gold-conjugated anti-GFP shows sparse but specific labelling in close proximity at the ciliary base and along the ciliary sheath at putative vesicles in WDR35::EmEGFP rescued *Wdr35* mutant MEFs as well as along the axoneme (see new Figure 8—figure supplement 2). It is important to note here that EmGFP epitopes were exposed to GFP antibodies by making 70nm thick serial sections of entire cell which explains the sparse but specific labelling. Cells are not permeabilised to preserve the membranes and vesicles. We continue to work to improve our strategies and protocols to better visualize sparse staining such as IFT-As whilst retaining both membrane and microtubule ultrastructures in mammalian cells.

3. I have some concerns about conclusions made based on ciliary localization and trafficking into the cilium in WDR35-/- cilia, in particular the fact that the Actub marker used for the cilia does not always stand out from other cytoplasmic microtubules. While I appreciate that >80% cells should have a short cilia in WDR35-/- it is not always clear where the cilia is in fluorescence imaging studies. There is considerable variability of Actub specific staining in cells. Issues with this and other figures are as follows. In Figure 4C for all 3 images representing WDR35 -/- conditions it is just not possible to identify precisely where the cilia is located, although it appears there is no accumulation of the lipid marker in any of the structures shown. There is a similar issue with Figure 3 Supplement 1 C in determining the length of the cilia being shown given the high non-ciliary signal observed for Actub. Figure 1D the authors indicate that the WDR35-/- cells are not polyglutamylated. It would be more convincing if another cilia marker was included since there is no quantification of the data provided or a centrosome marker was used. The conclusion reached about IFT-A components (Figure 3F) in WDR35-/- being in the transition zone seems hard to reach without the costaining with TZ markers or use of super resolution microscopy. In the case of Palmy-Myr-GFP localization for control and WDR35-/- cells, the later looks more like Golgi while the wt seems more vesicular. Thus, the authors conclusions about accumulation at the basal body of WDR35-/- is not well supported by the images shown in Figure 4C in my opinion. Overall, I find the fluorescence images could be better presented. In most cases too much space is taken up to show a field of view containing non-ciliary regions of the cell, while the centriole/cilia region are barely enlarged in cropped images. As quantification of some ciliary markers (ie TZ) was not done it might be better to show enlarged cropped areas of multiple cilia for representation.

Unfortunately for the quantification of cilia length in Figures 2A and 3F, as highlighted by reviewer 3, we are limited in mouse cells to availability of robust antibodies in a third species for colocalization studies (e.i. here we use AcTub (mouse) and IFT antibodies (rabbit)). Most available TZ markers were custom-made rabbit requested from other labs or no longer available (Santa Cruz large animals) aside from polyglutamylated tubulin (mouse), whose absence at the TZ is useful only if there is a marked axoneme on the other side which there is not in Wdr35 mutants. We have added in additional panels of cilia between genotypes focusing on the transition zone markers, as requested and reordered Figure 1 to enlarge these panels. Also without a TZ marker, we have altered the text to read ‘In contrast, in Wdr35^-/-^ MEFs, IFT-A core components fail to enter cilia and remain restricted at the ciliary base (Figure 3F),’ (line 225 of revised marked.docx) and edited similarly in the abstract. We also increased the panel sizes where possible for Figure 3F and Figure 4B, C. We changed the text summarizing localization of lipidated GFPs to read ‘In marked contrast to WT cells, in Wdr35^-/-^ MEFs, both the myristoylation and palmitoylation (MyrPalm) or dual palmitoylation (PalmPalm) EGFP failed to concentrate in mutant cilia (Figure 4C)’ (Line 390-401). We agree with the reviewer where these mislocalized cargos end up requires additional markers.

4. Given the number of full cell/cilia tomographs I wonder if the assertion that only 1% of cilia have fused with the plasma membrane is correct. In Video 4 it appears as though the tip of the cilium is possibly outside the cell. The percentage of outside cilia could be better evaluated by In and out assays (Kukic et al., 2016).

‘In and out’ assays would be a great way to validate this point but as it is very minor to our narrative, we have simply edited the text to read ‘very few (~1 %) cilia’ (line 427, marked revision.docx). (The ‘~’ was lost initially in my poor mastery of LaTeX and symbols). We looked at over 100 cells in these extensive analyses and our numbers are similar to what had been reported in the Molla-Herman paper for RPE-1 cells (Molla-Herman et al., 2010). The reviewer is very astute and correct indeed in Video 4 the last few frames are of a cilia tip emerging from the cell although the bulk of the axoneme is still internal, within the ciliary sheath.

[Editors' note: further revisions were suggested prior to acceptance, as described below.]

Reviewer #2:The authors have responded to my comments, particularly with respect to IFT peripheral subunits coating liposomes with specific lipid compositions. I do appreciate the novelty and the technical difficulties in carrying out TEM and CLEM experiments in the Wdr35 knockout cells and rescues, respectively. The additional Dynein 2 knockout TEM data is especially informative in suggesting that the accumulation of periciliary vesicles in Wdr35 knockout cells is unrelated to retrograde transport in cilia. However, I still have a few comments regarding these experiments and the interpretations intended for the general reader.First, the authors have mentioned in the rebuttal that the color coding for vesicles in Figure 7A is by genotype. However, the color code keys suggest otherwise. It is confusing as all vesicles in Wdr35 knockout are not "uncoated". The authors have clarified in the rebuttal that they would be uncomfortable showing the distribution of the different vesicle types, rather they quantify electron density around periciliary vesicles to show significant differences between genotypes. Thus, the terms "coated vesicles" and "coatless vesicles" in Figure 6, 7A and 9 are misleading and should be changed.

Although concerns about the usage of terms were not initially raised, in order to satisfy this reviewer, we have added the following statement to the Materials and methods section for Transmission Electron Microscopy for clarity on how they were assigned. ‘In wild type cells, very few vesicles are observed around the base of the cilia which mostly have electron-dense coats with a range of intensities, including those observed fusing with the ciliary pocket or sheath. For simplicity, we have labeled these as 'coated' and coloured them magenta. By comparison, the accumulation of vesicles around the base of Wdr35 mutant cilia generally lack prominent densities around them. For simplicity, we have labeled these as 'coatless' and coloured them cyan, as whether coats previously existed but disassembled or whether they failed to form in the first place is not clear from our study.’

Second, I might be wrong, but I don't see how the authors show by "inclusion of a wider field" in the tomographic studies of Wdr35 MEFs that the "vesicle accumulation is very regionalized over the 'thwarted' cilium". In the mentioned movies (Video 6, 7), I only see the predominant region around mother and daughter centrioles. To me, it is difficult to ascertain vesicles throughout an individual cell or in adjacent cells.

Video 7 corresponds to new Figure 6—figure supplement 3, which is made by stitching three sequential sections together and covers a whopping 9.3 µm * 9.3 µm field of view in XY and 900 nm in the Z-axis of three cells. To walk the reviewer through the events we capture in this volume, we provide the following description. Section1 (Z-STEP 0 to STEP 140) has a daughter centriole almost at the centre of the field of view and a few primarily coatless vesicles around its radius of 2 µm. Within this XY field of 9 µm * 9 µm, we do not see any vesicles away from the daughter centriole or close to or budding from the cell membrane in this section. Next section2 (Z-STEP 141 to STEP 264) has the basal body of the central cell. Again, although we see mostly coatless vesicles close to the basal body, no such vesicles are present at a distance from it. Coatless vesicles are also not seen close to the cell membrane or budding/fusing with the cell membrane. However, we did see clathrin-like vesicles with very pronounced electron densities close to and budding from the cell membrane. Three such vesicles are budding from the central cell are indicated by white arrows, with an additional one in the upper cell indicated by the black arrow in this section. Next, section-3 (Z-STEP 265 to STEP 420) has the transition zone of the central cell surrounded by mostly coatless vesicles in the periciliary space. Again, we do not see coatless vesicles beyond the 2 µm XY radius of the cilium, and none of these are budding from the cell membrane. This section also has clathrin-like electron-dense vesicles endocytosing from the cell membrane. One clathrin-like vesicle is seen budding from the central cell (white arrow), three from the upper cell (black arrow), and one from the third cell in the right (cyan arrow) in this section. In summary, this Video clearly shows the massive accumulation of mostly coatless vesicles in the periciliary space specifically and these accumulations are not found at a distance from it. We emphasize, that we have covered a very large volume of the cell while maintaining the resolution of our EM ultrastructure, which this reviewer fails to appreciate. To help explain this more clearly to our audience, we have now added these dimensions and edited the Figure 6—figure supplement 3 figure legend.

Third, quantification of periciliary vesicles, even from the limited number of cells observed from the single CLEM dataset as shown in Video 9, would really solidify the main message of the paper regarding accumulation of vesicles in the Wdr35 knockout.

We have quantified rescued cell vesicle numbers and fusion events in this revision as new Figure 8D. To clarify, Video 9, Figure 8C and Figure 8- supplement 1 where not from the CLEM dataset shown in Figure 8A, B rather similar conditions without the AiryScan imaging. We have edited Figure 8C and Figure 8—figure supplement 1 to help clarify which panels came from which experiment.

Fourth, given the limited data shown for CLEM for rescued cells in Figure 8A, the vesicles pointed as #1 and #2 in the CLEM data do not convincingly represent, at least to me, the vesicles pointed to in the adjacent EMs. A Video for the CLEM cell shown in Figure 8B can be included for clarity.

The CLEM dataset is very limited. There are a few 70nm sections for TEM available for the cell used for CLEM; we show six of the twelve serial micrographs available for this cell in Figure 8A (Z=1,3,4,8,9, and 10). Tomograms are unfortunately not available.

Fifth, for the "electron dense vesicles" #1 and #2 shown in Figure 8A-B and the three "coated" vesicles shown to be "fusing" (apparently from another rescued cell) in Figure 8C, how are the authors sure that they are not seeing clathrin-coated vesicles to be undergoing endocytosis? Clathrin coated vesicles endocytosing are also seen in Wdr35 knockout cells in Video 7. Are similar "fusions" seen in the wild type cells for the "coated" vesicles?

The reviewer is confused: 8C was Z sections through the same fusing vesicle, shown in full in Video 9. The cell shown in Video 9 and Figure 8C is different than cell used for CLEM in Figure 8A,B, which shows two coated vesicles at the base of another rescued cell. We spend significant time discussing why we do not believe these coated vesicles to be clathrin in the second paragraph of the discussion. Similar fusions are indeed seen in wild type cells (see Figure 6—figure supplement 1), and these fusion events are lacking in the mutants even when vesicles are closely apposed to the ciliary sheath see Figure 7 and Figure 6—figure supplement 2 (asterisks). We do not believe these are clathrin vesicles endocytosing because we can rescue them in *Wdr35* mutants by expressing WDR35, not clathrin or clathrin regulators. The reviewer is correct that we see several clathrin-like vesicles in the *Wdr35* null (Video 7), however these are always seen budding from the cell membrane and not the ciliary pocket. In contrast, in the wild type and rescued cells, we see coated vesicles fusing the ciliary pocket or sheath, and around the cilia base.

Finally, the new immunogold experiments with Wdr35-GFP are not very informative and need additional controls. As the authors mentioned in the text, Wolfrum et al. (2010) showed multiple IFT-A and IFT-B proteins to localize to vesicles adjacent to the photoreceptor connecting cilia. Other IFT proteins will probably show similar pattern in the context of primary cilia.

We are very disappointed this reviewer does not feel the addition of immunoEM is ‘very informative’. We have added controls including secondary only controls and non-transfected (thus expressing GFP) controls to highlight the specificity of the WDR35 immunogold signal, see Figure 8—figure supplement 3. We have also taken Reviewer 3’s suggestion to mark putative vesicles as pink outlines as opposed to objects, as done in the Sedmak and Wolfrum 2010 paper.